# *Pb*AP2-FG2 and *Pb*AP2R-2 function together as a transcriptional repressor complex essential for *Plasmodium* female development

**Tsubasa Nishi[1], Izumi Kaneko[1], Shiroh Iwanaga[2], Masao Yuda[1]\***

**1** Laboratory of Medical Zoology, Department of Medicine, Mie University, Tsu, Japan, **2** Department of Molecular Protozoology, Research Institute for Microbial Diseases, Osaka University, Suita, Japan

\* m-yuda@med.mie-u.ac.jp

**Data Availability Statement:** The authors confirm that all data underlying the findings are fully available without restriction. All fastq files for ChIP-seq and RNA-seq experiments are available from

## Abstract

Gametocyte development is a critical step in the life cycle of *Plasmodium*. Despite the number of studies on gametocyte development that have been conducted, the molecular mechanisms regulating this process remain to be fully understood. This study investigates the functional roles of two female-specific transcriptional regulators, *Pb*AP2-FG2 and *Pb*AP2R-2, in *P. berghei*. Knockout of *pbap2-fg2* or *pbap2r-2* impairs female gametocyte development, resulting in developmental arrest during ookinete development. ChIP-seq analyses of these two factors indicated their colocalization on the genome, suggesting that they function as a complex. These analyses also revealed that their target genes contained a variety of genes, including both male and female-enriched genes. Moreover, differential expression analyses showed that these target genes were upregulated through the disruption of *pbap2-fg2* or *pbap2r-2*, indicating that these two factors function as a transcriptional repressor complex in female gametocytes. Formation of a complex between *Pb*AP2-FG2 and *Pb*AP2R-2 was confirmed by RIME, a method that combines ChIP and MS analysis. In addition, the analysis identified a chromatin regulator *Pb*MORC as an interaction partner of *Pb*AP2-FG2. Comparative target analysis between *Pb*AP2-FG2 and *Pb*AP2-G demonstrated a significant overlap between their target genes, suggesting that repression of early gametocyte genes activated by *Pb*AP2-G is one of the key roles for this female transcriptional repressor complex. Our results indicate that the *Pb*AP2-FG2-*Pb*AP2R-2 complex-mediated repression of the target genes supports the female differentiation from early gametocytes.

## Author summary

Gametocyte development in *Plasmodium* parasites, a causative agent of malaria, is an essential step for their transmission from vertebrate hosts to mosquitoes. Gametocytes are sexual precursor cells produced from a subpopulation of asexual blood-stage parasites. Upon uptake by mosquitoes through blood feeding, the male and female gametocytes become microgametes and macrogametes, respectively, and then they fertilize and develop into the mosquito midgut invasive stage, called ookinete. Therefore, it is crucial to understand the underlying mechanisms regulating this developmental process. This

the GEO database (accession numbers GSE198588, GSE213776).

**Funding:** This work was supported by the Japan Society for the Promotion of Science (17H01542 to YM; 20K07462 to IK; 21K06986 to TN). The funders had no role in study design, data collection and analysis, decision to publish, or preparation of the manuscript.

**Competing interests:** The authors have declared that no competing interests exist.

study revealed that the two female transcriptional regulators, *Pb*AP2-FG2 and *Pb*AP2R-2, function together as an essential transcriptional repressor complex in *P. berghei*, the target genes of which include male, female, and early gametocyte genes activated by *Pb*AP2-G. Our findings suggest that *Pb*AP2-FG2 and *Pb*AP2R-2 play multiple roles in supporting the development of female gametocytes from early gametocytes.

## Introduction

*Plasmodium* parasites are the causative agent of malaria, one of the most severe infectious diseases worldwide. The spread of the parasites among individuals occurs through mosquito bites, resulting in more than 200 million cases and 500 thousand deaths yearly [1]. Parasite transmission from vertebrate hosts to mosquitoes is involved in the sexual development of the parasite [2,3]. During asexual reproduction in the host blood, subpopulations of parasites differentiate into gametocytes to prepare for transmission to mosquitoes [4,5]. When the gametocytes are taken up by mosquitoes through blood feeding, they egress from red blood cells, form gametes, and fertilize. The fertilized cells then develop into ookinetes and invade the midgut of mosquitoes, completing the transmission [6]. Parasite transmission is a critical event in the propagation of malaria, and thus understanding the molecular mechanisms regulating these developmental steps is crucial for malaria epidemiology.

Along with several transcriptomic [7–10] and proteomic studies [11,12], studies of gametocyte-specific transcriptional regulators have been conducted to elucidate the mechanisms of *Plasmodium* gametocyte development. In *Plasmodium* spp., sexual development is triggered by AP2-G [13–15]. It is an AP2-family transcription factor that is expressed in a subpopulation of blood-stage parasites, and disruption of the gene results in complete loss of the parasite's ability to produce gametocytes [16,17]. Furthermore, it has been reported that intentional conversion of parasites into the sexual stage could be achieved by conditional induction of AP2-G in both *P. falciparum* and *P. berghei* [18,19]. We previously conducted chromatin immunoprecipitation (ChIP) followed by high-throughput sequencing (ChIP-seq) analysis of *Pb*AP2-G and identified its target genes [20]. Among these targets of *Pb*AP2-G, we found several important transcription factors for sexual development. These targets include *pbap2-g2*, a transcription factor that is expressed in both male and female gametocytes. In both *P. berghei* and *P. falciparum*, AP2-G2 functions as a transcriptional repressor to ensure the alteration of cell fate from the asexual blood stage to the sexual stage [21,22]. The target genes of *Pb*AP2-G also include a female-specific transcription factor gene, *pbap2-fg* [23,24]. *Pb*AP2-FG activates most female-enriched genes, and disruption of this gene results in the formation of abnormal female gametocytes [23]. In addition to these gametocyte-specific transcription factors, the zygote transcription factor gene *pbap2-z* was also identified as a target of *Pb*AP2-G [25]. Moreover, a transcriptional regulator gene, *pfhdp1*, that is essential for early gametocyte development was included in the target genes of *Pf*AP2-G, and its ortholog was also a target of *Pb*AP2-G [26,27]. Therefore, it was considered that *Pb*AP2-G comprehensively activates transcriptional regulator genes essential for *Plasmodium* sexual development.

*pbap2-o3* and *pbap2r-2* are also a target gene of *Pb*AP2-G. It has been reported that in both *P. berghei* and *P. yoelii*, disruption of *ap2-o3* results in arrest of parasite development during ookinete development, thereby helping derive the name *ap2-o3* [28,29]. *Pb*AP2R-2, on the other hand, is expressed in female gametocytes, and disruption of this gene impairs ookinete development [20]. Here, we report that *Pb*AP2-O3 and *Pb*AP2R-2 are expressed in female gametocytes and function together as a transcriptional repressor complex in *P. berghei*.

Accordingly, we renamed *Pb*AP2-O3 *Pb*AP2-FG2. Moreover, differential expression analyses and ChIP-seq analyses revealed that *Pb*AP2-FG2 and *Pb*AP2R-2 repress various genes, including female, male, and early gametocyte genes, to support female development. Recently, Li *et al.* reported that the ortholog of *Pb*AP2-FG2 in another rodent malaria parasite, *P. yoelii*, (*Py*AP2-O3) also functions as a transcriptional repressor in female gametocytes [30]. Although some of their conclusions were inconsistent with those in this study, reassessment of their data led to similar conclusions in both *P. berghei* and *P. yoelii*.

## Results

### *Pb*AP2-FG2 is expressed in female gametocytes and is essential for their development

*Pb*AP2-FG2 (encoded by PBANKA_1015500), previously named *Pb*AP2-O3, is an AP2-family transcription factor, conserved across the *Plasmodium* species. It possesses a single AP2 domain near its N-terminus and an AP2-coincident domain mostly at the C-terminus (ACDC) domain (Fig 1A). Modrzynska *et al.* demonstrated that disruption of this gene results in the failure of the majority of zygotes to form an apical protrusion in *P. berghei* [28]. This result indicated that *Pb*AP2-FG2 has a role in zygote/ookinete development or regulating female transcriptome because female gametocytes store large amount of mRNAs that are essential for development after fertilization. Our previous ChIP-seq data demonstrated that the upstream region of *pbap2-fg2* harbored binding sites of *Pb*AP2-G and *Pb*AP2-FG (encoded by PBANKA_1437500 and PBANKA_1415700, respectively), which are transcriptional activators expressed in early gametocytes and female gametocytes, respectively (S1 Fig) [20,23]. This indicates that *Pb*AP2-FG2 is possibly expressed in female gametocytes. To identify the stage at which *Pb*AP2-FG2 functions, we first generated a parasite line expressing GFP-fused *Pb*AP2-FG2 (*Pb*AP2-FG2::GFP, S2A Fig) and assessed its expression pattern. In the blood stage, *Pb*AP2-FG2 expression was observed in the nucleus of female gametocytes but not in the other stages including male gametocytes (Fig 1B). When cultured in an ookinete culture medium, *Pb*AP2-FG2::GFP parasites produced banana-shaped ookinetes, confirming that the GFP fusion did not impair *Pb*AP2-FG2 function. During ookinete development, no fluorescence was detected at any stage of the parasites (Fig 1B). These results collectively indicated that like *Py*AP2-O3 (encoded by PY17X_1017000), *Pb*AP2-FG2 is only expressed in female gametocytes during sexual development and presumably functions in females instead of zygotes.

To investigate the function of *Pb*AP2-FG2, we developed *pbap2-fg2* knockout parasites [*pbap2-fg2*(-), S2B Fig] and evaluated the phenotype in detail. *pbap2-fg2* was disrupted by double cross-over homologous recombination of an *hdhfr* expression cassette, with the transfection conducted twice to obtain two independent clonal lines. For both of the clones, the resultant parasites formed morphologically normal female and male gametocytes (Fig 1C), and the male gametocytes showed normal exflagellation (Fig 1D). In the ookinete culture medium, *pbap2-fg2*(-) parasites failed to produce banana-shaped ookinetes; more than half of the fertilized population stopped developing at round zygotes, and the others, at retort-form ookinetes (Fig 1E). Furthermore, both of the clones failed to infect mosquitoes through blood feeding (Fig 1F), confirming that *pbap2-fg2*(-) parasites completely lost the ability to produce normal ookinetes. These results corroborated the findings of Modrzynska *et al.* [28]. Next, we performed a cross-fertilization assay using the clone 1 to evaluate whether female or male gametocytes of *pbap2-fg2*(-) were capable of forming banana-shaped ookinetes upon fertilization with normal gametocytes. We observed that crossing of *pbap2-fg2*(-) with a line that produced infertile females [*p47*(-)] [31] led to no female gametocytes being converted to banana-shaped

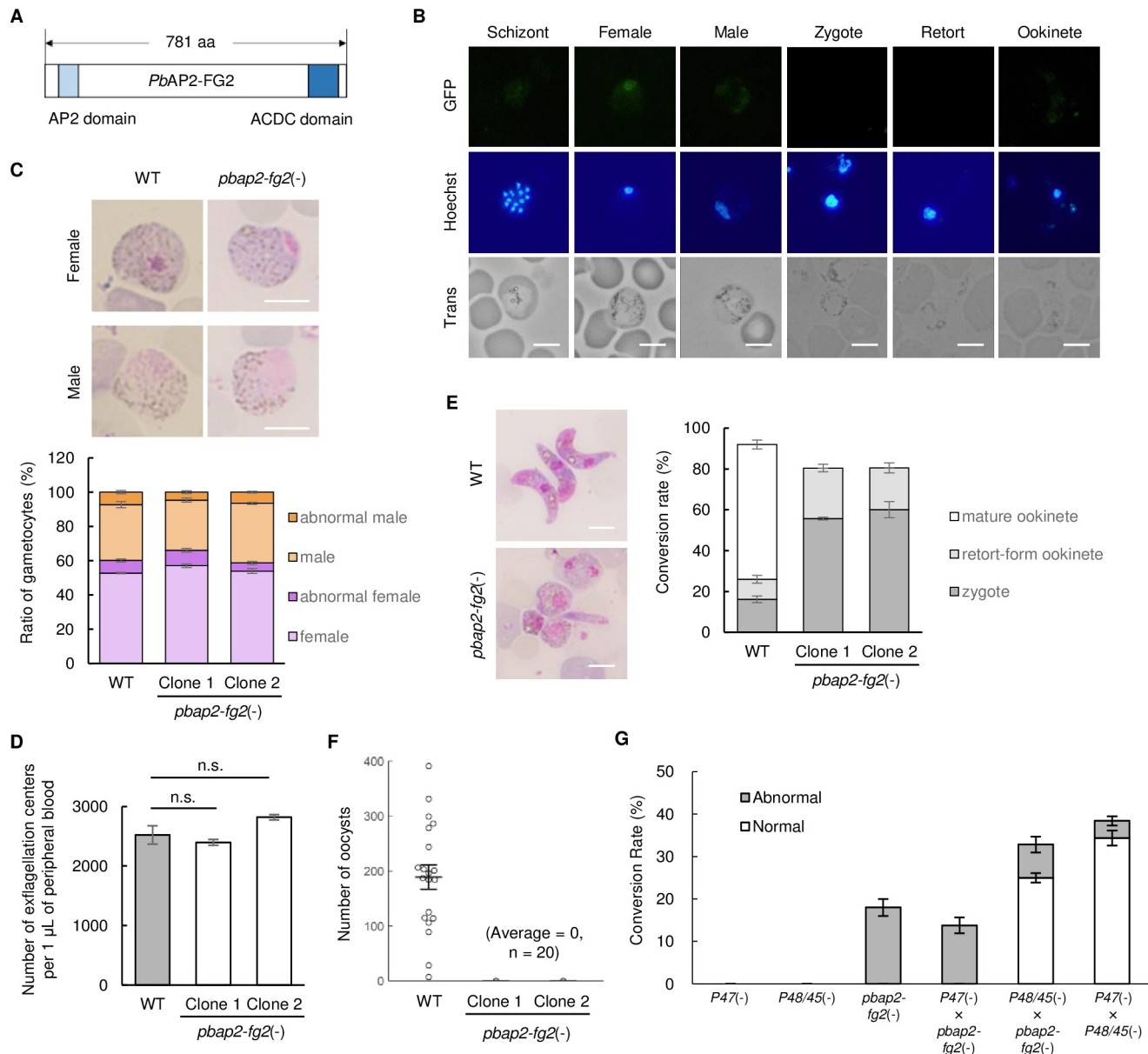

**Fig 1. Expression pattern and knockout phenotype of *pbap2-fg2*.** (A) Schematic diagram of protein features in *Pb*AP2-FG2. (B) Fluorescence analysis of *Pb*AP2-FG2::GFP during blood-stage and sexual development. Nuclei were stained with Hoechst 33342. Scale bar = 5 μm. (C) Top: representative Giemsa-stained images showing gametocytes of WT and *pbap2-fg2*(-) parasites. Scale bar = 5 μm. Bottom: ratio of normal/abnormal female and male gametocytes in WT and *pbap2-fg2*(-). Error bars indicate the standard error of the mean (n = 3). (D) Number of exflagellation centers per 1 μL of peripheral blood from mice infected with WT and *pbap2-fg2*(-). The infected mice were treated with sulfadiazine in their drinking water to enrich gametocyte population in their blood stream. Error bars indicate the standard error of the mean (n = 3). (E) Left: representative Giemsa-stained images showing ookinetes of WT and *pbap2-fg2*(-) at 20 h after starting ookinete cultures. Scale bar = 5 μm. Right: Rate of conversion to zygote, retort-form ookinete, and mature ookinete against all female-derived cells in WT and *pbap2-fg2*(-) at 20 hoc. Zygote and retort-form ookinete include morphologically abnormal cells without and with an apical protrusion, respectively. Error bars indicate the standard error of the mean (n = 3). (F) Number of midgut oocysts at 14 days post infection for WT and *pbap2-fg2*(-). Lines indicate the mean values and the standard error (n = 20). (G) Cross-fertilization assay among *pbap2-fg2*(-), *p48/45*(-) and *p47*(-). The number of normal and abnormal ookinetes are indicated as white and grey bars, respectively. Error bars indicate the standard error of the mean (n = 3).

ookinetes (Fig 1G). Development of their fertilized females was arrested at the round zygote or retort-form ookinete, recapitulating the scenario of *pbap2-fg2*(-) parasites cultured alone. In contrast, when *pbap2-fg2*(-) was crossed with a line that produces infertile males [*p48/45*(-)]

[32], approximately 30% of female gametocytes were converted to banana-shaped ookinetes, which was approximately as much as when *p48/45*(-) and *p47*(-) were crossed (Fig 1G), demonstrating the ability of *pbap2-fg2*(-) to produce normal male gametocytes. Collectively, these results revealed that only female gametocytes were abnormal in *pbap2-fg2*(-), which, in turn, affected their ookinete development. Together with the fluorescence analysis, these results strongly suggested that *Pb*AP2-FG2 is essential for the development of normal female gametocytes, which is consistent with the study of the *Py*AP2-O3 reported by Li *et al.* [30].

## Disruption of *pbap2-fg2* affected the female transcriptome

To further investigate the effect of disrupting *pbap2-fg2* on female development, we performed RNA-seq analysis on gametocyte-enriched populations of wild-type ANKA strain (WT) and *pbap2-fg2*(-), and compared their transcriptomes. The total RNA was harvested from parasites enriched with gametocytes, which were prepared by killing asexual parasites with sulfadiazine treatment, and then sequenced using next-generation sequencing. Differentially expressed genes (DEGs) between WT and *pbap2-fg2*(-) were identified by analyzing the sequence data using DESeq2 after excluding genes with reads per kilobase of transcript per million mapped reads (RPKM) < 10 (S1A Table). In *pbap2-fg2*(-) parasites, 180 genes were significantly downregulated [$\log_2$(fold change) < -1, *p*-value adjusted for multiple testing with the Benjamini-Hochberg procedure (*p*-value adj) < 0.05], and 96 genes were significantly upregulated [$\log_2$(fold change) > 1, *p*-value adj < 0.05] compared to the WT (Fig 2A, S1B and S1C Table). To evaluate how disruption of *pbap2-fg2* affects gametocyte transcriptome, we assessed the expression of these DEGs in previously reported sex-specific RNA-seq data

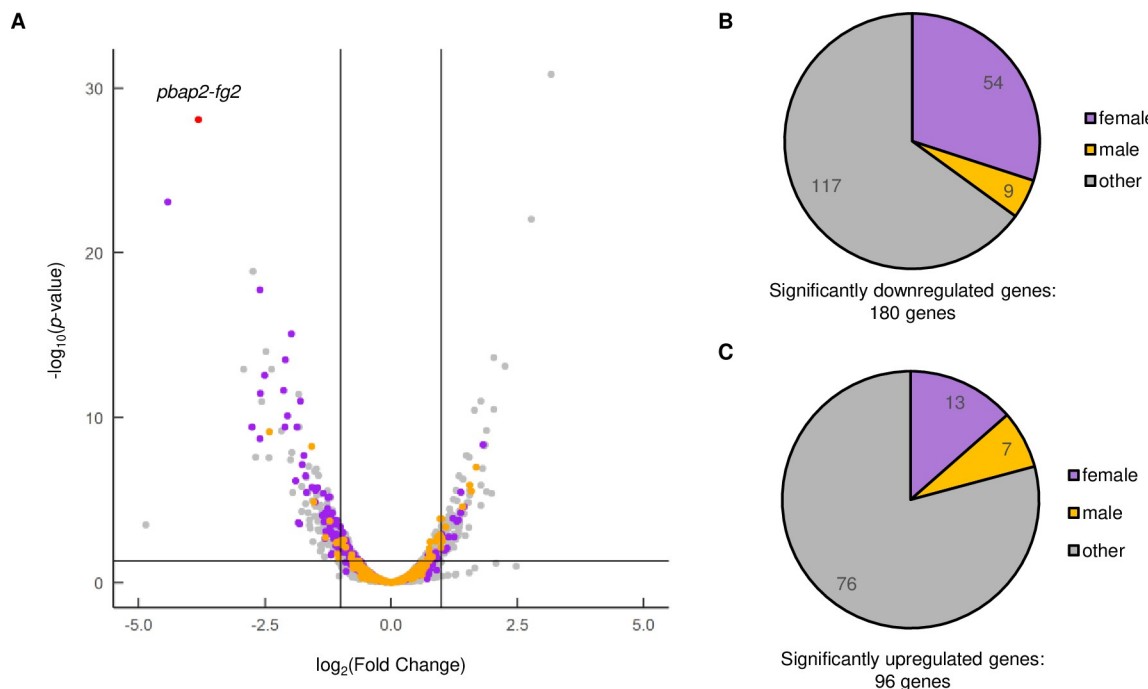

**Fig 2. Differential expression analysis between WT and *pbap2-fg2*(-).** (A) A volcano plot showing differential expression of genes in *pbap2-fg2*(-) compared to WT. Purple and orange dots represent female and male-enriched genes, respectively. A red dot indicates *pbap2-fg2*. A horizontal line indicates *p*-value of 0.05, and two vertical lines indicate $\log_2$(Fold Change) of -1 and 1. (B) Classification of significantly downregulated genes into sexual stage-enriched gene sets. All genes not enriched in either "female" or "male" are classified as "other". (C) Classification of significantly upregulated genes into sexual stage-enriched gene sets.

[10]. We identified genes more than fourfold enriched with *p*-value adj < 0.001 in each sexual stage compared to the other and asexual blood stages as sex-enriched genes and obtained 504 female-enriched genes and 438 male-enriched genes (all the other genes were assigned as "other" in this context) (S2A and S2B Table). The genes downregulated in *pbap2-fg2*(-) contained 54 female-enriched genes and nine male-enriched genes, highlighting the dominance of female-enriched genes with *p*-value = $7.0 \times 10^{-14}$ by Fisher's exact test (Fig 2A and 2B). Furthermore, in the total transcriptome, log$_2$(fold change) of female-enriched genes tended to be lower than the other genes (*p*-value = $1.4 \times 10^{-18}$ by two-tailed Student's t-test). In contrast, the upregulated genes showed no specific enrichment in the female- or male-enriched genes (Fig 2A and 2C). These results indicated that disruption of *pbap2-fg2* impaired the female transcriptome, causing downregulation of female-enriched genes.

## *Pb*AP2-FG2 targets a wide variety of genes, binding to specific sequences

Differential expression analysis between WT and *pbap2-fg2*(-) suggested that *Pb*AP2-FG2 was involved in transcriptional regulation in female gametocytes. Therefore, we employed ChIP-seq analysis to identify the binding motif of *Pb*AP2-FG2 and its target genes. We performed ChIP experiments with *Pb*AP2-FG2::GFP using an anti-GFP antibody, followed by the sequencing of the DNA fragments purified from the immunoprecipitated chromatin and the input cell lysate. From the sequence data, peaks were called with fold enrichment > 3.0 and *q*-value < 0.01 using the macs2 program, setting the sequence data of input DNA as a control. Two biologically independent experiments were performed, the results of which were comparable based on the genome-wide peak pattern of the data (Fig 3A). We identified 1321 and 1648 peaks in Experiments 1 and 2, respectively, and the locations of 1231 peaks (93.2% of the peaks from Experiment 1) overlapped between the two experiments, suggesting that the data had high reproducibility. To further evaluate the reproducibility of each peak, IDR1D analysis was performed on the two data [33,34]. In this analysis, peaks were ranked according to their *p*-values within each replicate, and the ranks were compared between the two experiments. According to the consistency of ranks across the two experiments, the irreproducible discovery rate (IDR) score, which defines the reproducibility of each peak, was calculated for each one. As the ranks of each peak lose consistency between the replicates, the peaks have higher IDR scores; hence, peaks with small IDR scores are considered reliable. The results depicted that 638 peaks had an IDR < 0.01 (Fig 3B, S3A and S3B Table), and we decided to utilize these peaks for further analysis as they are highly reproducible.

We first attempted to identify the binding motif of *Pb*AP2-FG2 by searching for statistically enriched sequences around the highly reproducible peaks. We searched for 6-bp motifs on sequences within 100 bp from each summit and found enrichment of several motifs using Fisher's exact test. These motifs were unified, and RGAGAR (R = A or G) was identified as the most significantly enriched motif in the peak regions with a *p*-value of $5.2 \times 10^{-89}$ (Fig 3C). In addition to the RGAGAR motifs, we found GAGARA and ARGAGA as enriched motifs with a *p*-value of $2.6 \times 10^{-68}$ and $3.5 \times 10^{-53}$, respectively (Fig 3C). These motifs appeared to be variants of the most enriched motif, sharing GAGA within their sequences. Accordingly, we hereafter refer to the RGAGAR motif as the major motif and the GAGARA and ARGAGA motifs as the variant motifs 1 and 2, respectively. Searching for these three motifs around peak summits revealed that 85% of the peaks had at least one of these enriched motifs within 300 bp of the summit (Fig 3C). Moreover, for more than half of these peaks, the distance between the peak summit and the nearest motif was within 50 bp (Fig 3D). These results indicated that *Pb*AP2-FG2 binds to the major motif RGAGAR and its variant motifs.

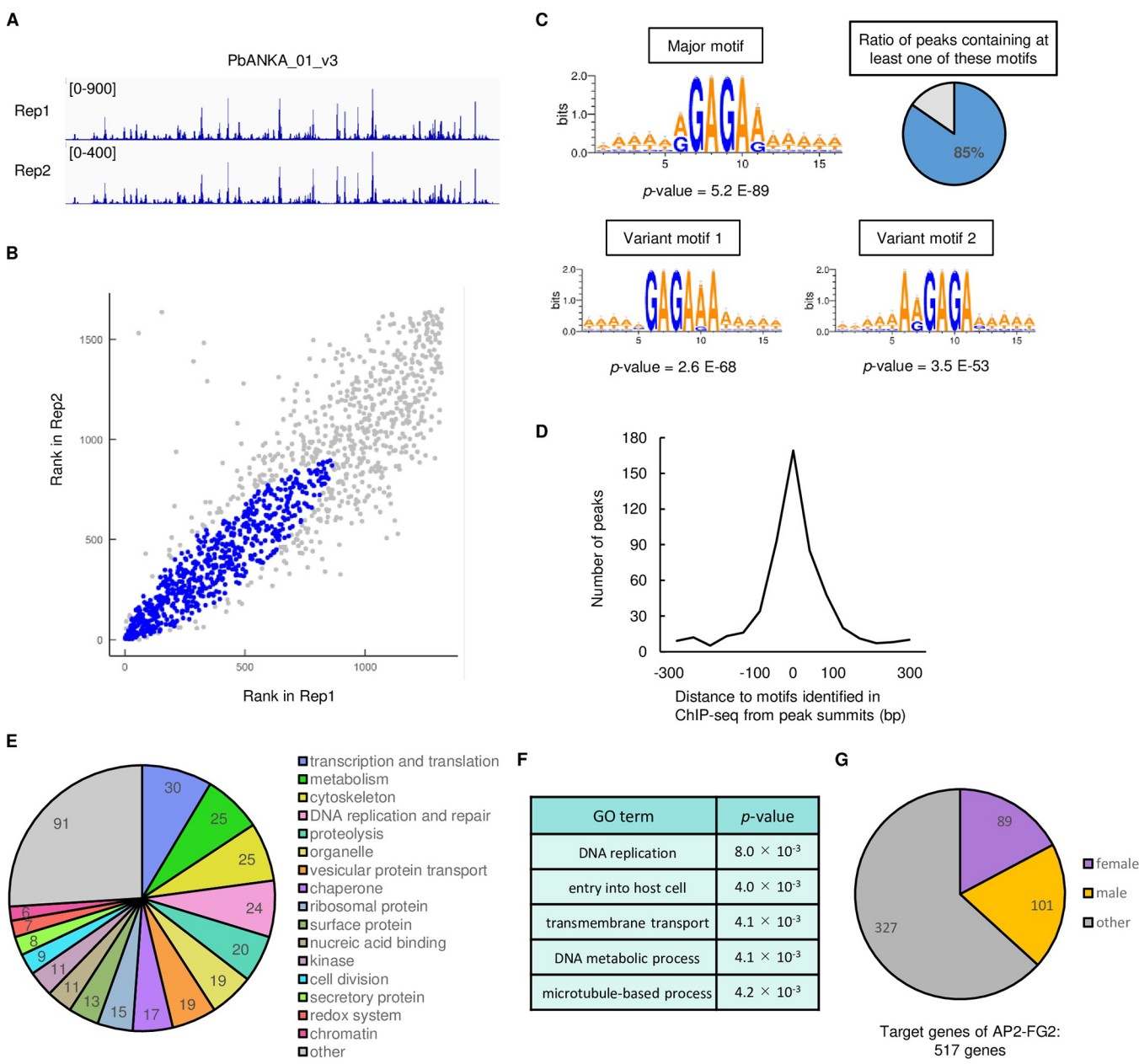

**Fig 3. ChIP-seq analysis using *Pb*AP2-FG2::GFP.** (A) IGV images showing peaks identified in the ChIP-seq experiment 1 and 2 of *Pb*AP2-FG2 on the chromosome 1. Read coverage for the ChIP data is shown. (B) IDR1D analysis between the ChIP-seq experiment 1 and 2. The rank of peaks according to their *p*-value for each experiment is plotted against each other. Peaks with IDR < 0.01 are indicated as blue dots. (C) Motifs enriched within 50 bp from peak summits identified in the ChIP-seq of *Pb*AP2-FG2. The logos were depicted using WebLogo 3 (http://weblogo.threeplusone.com/). Percentage of the peaks that had at least one of these motifs within 300 bp from the summit is indicated as a pie graph in the top left corner. (D) Distance between peak summits and the nearest major or variant motifs. (E) Classification of target genes into 17 groups according to their functional annotation. (F) Gene ontology analysis for target genes of *Pb*AP2-FG2. Terms with *p*-value < 0.01 are shown. (G) Classification of target genes of *Pb*AP2-FG2 into sexual stage-enriched gene sets.

Next, we analyzed the genomic location of the peaks identified by ChIP-seq analysis to determine the potential targets of *Pb*AP2-FG2. The analysis revealed that of the 638 highly reproducible peaks, 558 peaks located in intergenic regions. These intergenic peaks located in the upstream regions (within 1200 bp from ATG) of 517 genes, suggesting that these genes are a target gene of *Pb*AP2-FG2 (S3C Table). Of the 517 target genes, 350 have been functionally

annotated on PlasmoDB (https://plasmodb.org). We classified these 350 genes into functional groups to evaluate functional characteristics of the target genes (Fig 3E and S3C Table). The target genes contained some groups that included many genes highly transcribed in female gametocytes, such as "cytoskeleton" and "secretory protein". The group "cytoskeleton" had some genes encoding inner membrane complex and myosin proteins [35,36], and the group "secretory protein" contained *warp* and some secreted ookinete protein genes [37,38]. Meanwhile, some other functional groups, *such as* "DNA replication and repair" and "cell division", did not seem to be related to female development. To further investigate whether genes of any specific function were enriched in the targets, we performed a gene ontology (GO) analysis. The GO analysis revealed that the target genes were most enriched in the term "DNA replication," which included putative DNA replication licensing factor genes, DNA polymerase subunit genes and so on, with $p$-value of $8.0 \times 10^{-4}$ (Fig 3F). In addition, genes that belong to the GO terms "entry into host cell," "transmembrane transport," "DNA metabolic process," and "microtubule-based process" were also found to be enriched ($p$-value $< 0.01$) (Fig 3F).

Next, we evaluated the composition of sex-enriched genes among the targets of *Pb*AP2-FG2. Of the 517 target genes, 90 were female-enriched (Fig 3G and S3C Table). However, in these female-enriched target genes, genes of any specific function did not appear enriched. The targets also contained 101 male-enriched genes (Fig 3G and S3C Table), including most of the genes that were classified into the functional groups "DNA replication and repair" and "cell division". Because *Pb*AP2-FG2 is a female-specific transcription factor, the considerable number of male-enriched genes in the targets may imply the possible role of *Pb*AP2-FG2 as a transcriptional repressor. Concordantly, when the association between the target genes of *Pb*AP2-FG2 and genes downregulated in *pbap2-fg2*(-) was assessed, only four target genes were significantly downregulated. Therefore, we considered that the downregulation of genes in *pbap2-fg2*(-) was not a direct effect of disrupting *pbap2-fg2*.

## Target genes of *Pb*AP2-FG2 were upregulated in *pbap2-fg2*(-)

To evaluate how the disruption of *Pb*AP2-FG2 affected the transcription of its target genes, we compared the target genes and DEGs identified in the RNA-seq analysis. Intriguingly, the target genes were enriched in the genes significantly upregulated in *pbap2-fg2*(-) (40 of the 96 upregulated genes, $p$-value $= 6.9 \times 10^{-15}$ by Fisher's exact test), while only four targets were included in the significantly downregulated genes, as mentioned above (Fig 4A and S3C Table). In addition, although not more than 2-fold, the other 47 target genes were upregulated, with a $p$-value adj $< 0.05$. Furthermore, comparing the $\log_2$(fold change) distribution of the target genes with that of the other genes revealed that the target genes tended to be upregulated in *pbap2-fg2*(-) with a $p$-value of $8.2 \times 10^{-58}$ by two-tailed Student's t-test (Fig 4A). Therefore, we considered that *Pb*AP2-FG2 repressed the transcription of its target genes in female gametocytes. An upregulation pattern of the target genes was also observed in female and male-enriched genes with a $p$-value of $3.9 \times 10^{-13}$ and $1.2 \times 10^{-4}$ by two-tailed Student's t-test, respectively (Fig 4B and 4C), indicating a role of *Pb*AP2-FG2 in repressing its target genes regardless of their sex-enriched expression property.

Considering the above results, we examined whether the binding motifs of *Pb*AP2-FG2 were enriched in the upstream region (300–1200 bp from ATG) of the upregulated genes compared to that of the other genes. Through this analysis, we found that one of the motifs that belong to the major motifs, GGAGAG, was found to be the most enriched by Fisher's exact test ($p$-value $= 4.4 \times 10^{-6}$, Fig 4D). Additionally, two other major motifs (CTCTCT and CTCTCC) and one of the variant motifs 1 and 2 (TTTCTC and AGGAGA, respectively) were also found to be enriched, with a $p$-value $< 0.005$ (Fig 4D). These results strongly suggested that the upregulation of genes in *pbap2-fg2*(-) was primarily a direct effect of its disruption. In

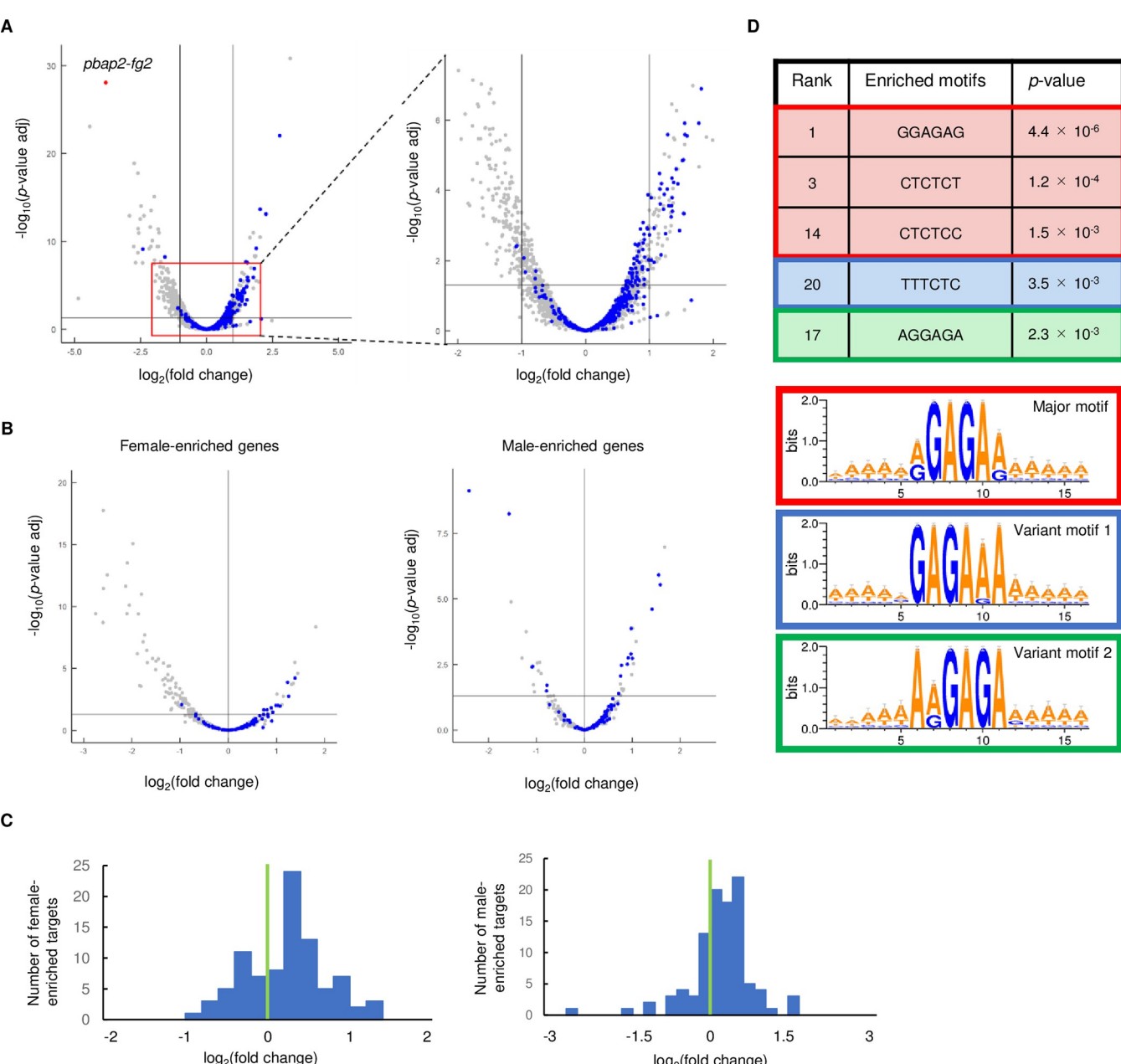

**Fig 4. Relationship between target genes of *Pb*AP2-FG2 and DEGs in *pbap2-fg2*(-).** (A) A volcano plot showing DEGs in *pbap2-fg2*(-) compared to WT. Blue dots represent the target genes of *Pb*AP2-FG2, and a red dot indicates *pbap2-fg2*. A horizontal line indicates *p*-value of 0.05, and two vertical lines indicate $\log_2$(Fold Change) of -1 and 1. All genes with RPKM $\geq$ 10 are depicted in the left panel. For the right panel, region from $\log_2$(Fold Change) of -2 to 2 and from $-\log_{10}$(*p*-value) of 0 to 7.5 is magnified. (B) Volcano plots showing DEGs in *pbap2-fg2*(-) for female and male-enriched genes (the left and right panel, respectively). Blue dots represent the target genes of *Pb*AP2-FG2. A horizontal line indicates a *p*-value of 0.05, and a vertical line indicates $\log_2$(Fold Change) of 0. (C) Distribution of $\log_2$(Fold Change) values for female and male-enriched target genes of *Pb*AP2-FG2 (the left and right graph, respectively). A green line indicates $\log_2$(Fold Change) of 0. (D) Six-bp DNA motifs enriched within the upstream region (300 to 1200 bp from ATG) of genes upregulated in *pbap2-fg2* (-). The major and variant motifs are each indicated in different color boxes. The ranks were assigned to all enriched motifs according to their *p*-values.

addition, this comparison of the differential expression analysis and target analysis indicated the importance of target analysis by ChIP-seq for identifying direct targets of a transcriptional regulator because it is difficult to distinguish direct and indirect effects resulted from a gene knockout by RNA-seq analysis alone.

## The binding motifs of *Pb*AP2-FG2 functioned as a *cis*-acting repressive element

To confirm that *Pb*AP2-FG2 functions as a transcriptional repressor upon binding to the major and variant motifs, we evaluated the effects of disrupting motifs in the upstream region of target genes. First, we generated parasites expressing GFP-fused *Pb*AP2-FG2 by the CRISPR/Cas9 system (*Pb*AP2-FG2::GFP[C], S2C Fig) using Cas9-expressing parasites called Pbcas9 [39]. After developing *Pb*AP2-FG2::GFP[C], we introduced point mutations in the motif upstream of *psh3* (PBANKA_1223500). Parasite-specific helicase 3 (PSH3) is a helicase conserved in apicomplexan parasites, which in *P. falciparum* is reported to be expressed in trophozoites and schizonts and is essential for asexual stage development [40]. In addition, *psh3* is also reported to be essential for asexual development of *P. berghei* in the PlasmoGEM study [41]. The female and male gametocyte RNA-seq data indicated that *psh3* is transcribed in male but not female gametocytes. In the differential expression analysis between WT and *pbap2-fg2* (-), *psh3* was significantly upregulated in *pbap2-fg2*(-) with $\log_2$(fold change) of 2.0, assuring that it is a true target gene of *Pb*AP2-FG2. Using the CRISPR/Cas9 system, we changed the motif in the peak region upstream of *psh3* from GGAGAA to atAtAt (Fig 5A).

To evaluate the binding of *Pb*AP2-FG2 at the mutated site, we first performed ChIP coupled with quantitative PCR (ChIP-qPCR) analysis using *Pb*AP2-FG2::GFP[C] with the wild-type (motif_WT) or mutated (motif_mutated) motif upstream of *psh3*. In the motif_mutated parasites, the amount of immunoprecipitated DNA fragments relative to input DNA (%input

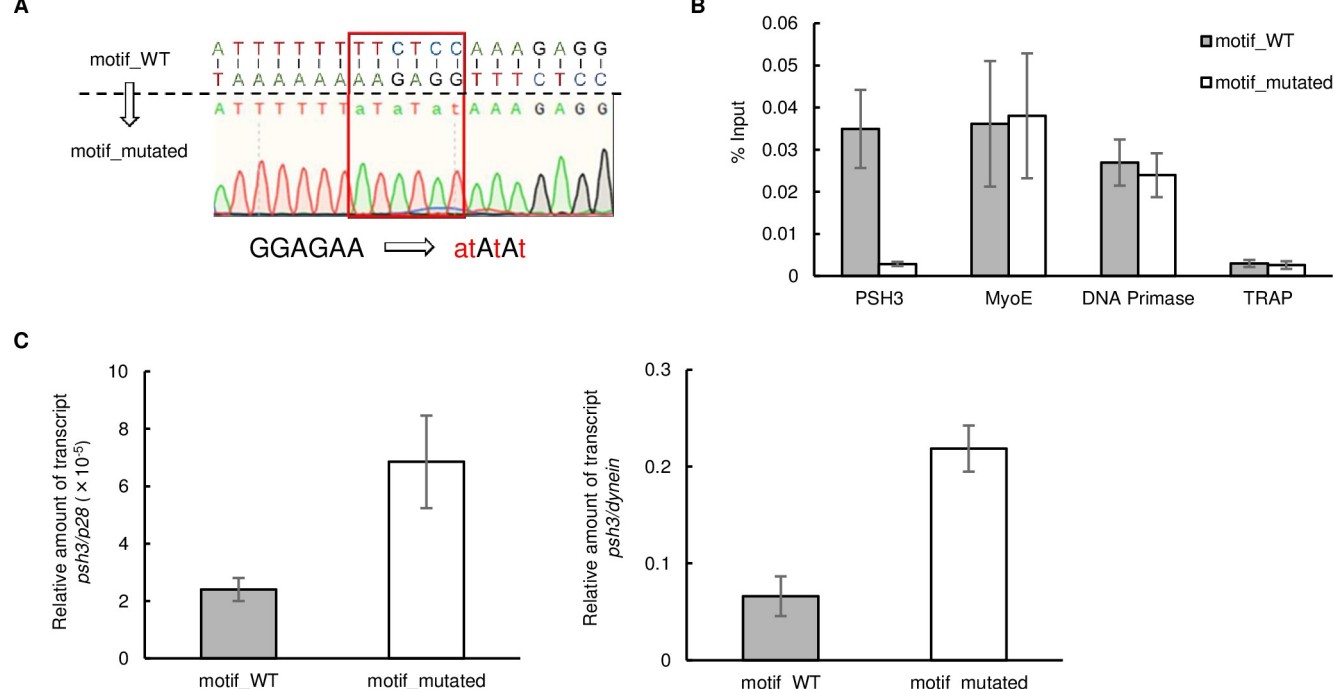

**Fig 5. Disruption of the binding motif of *Pb*AP2-FG2 in the upstream region of *psh3*.** (A) Genomic sequence around the binding motif of *Pb*AP2-FG2 located upstream of *psh3*, and Sanger sequence result of the region in motif_mutated parasites. (B) ChIP-qPCR analysis of *Pb*AP2-FG2 at the mutated site upstream of *psh3*. Grey and white bars indicate %input for motif_WT and motif_mutated, respectively. Error bars indicate the standard error of the mean % input values from three independent experiments. [PSH3: Parasite-Specific Helicase 3, MyoE: Myosin E, DNA Primase: DNA Primase large subunit, TRAP: Thrombospondin-Related Anonymous Protein] (C) RT-qPCR analysis of *psh3* in motif_WT and motif_mutated parasites. The relative transcript level of *psh3* against *p28* and *dynein* is presented at the left and right, respectively. Error bars indicate the standard error of the mean from three independent experiments.

value) was significantly decreased at the mutated site compared to the motif_WT (Fig 5B). In contrast, at the other sites, the upstream region of *myoE* and the DNA primase large subunit gene as positive controls and *trap* as a negative control, the %input values were comparable between the motif_WT and motif_mutated parasites (Fig 5B). Together with the ChIP-seq results, these results strongly indicated that the ChIP-seq-identified motifs are the binding motifs of *Pb*AP2-FG2.

Subsequently, using these mutants, we assessed how the binding of *Pb*AP2-FG2 affects downstream transcription by reverse transcription quantitative PCR (RT-qPCR) analysis. Total RNA was harvested from motif_WT and motif_mutated parasites treated with sulfadiazine, and the relative amount of *psh3* transcripts to *p28* transcripts was analyzed. The results demonstrated that the relative transcript level of *psh3* in motif_mutated parasites was more than 2.5-fold higher than in motif_WT (Fig 5C, left graph). Since *psh3* is usually expressed in male gametocytes but not in females, we supposed that it is essential to exclude the possibility that the male-to-female ratio affected the result. Accordingly, we evaluated the amount of *psh3* transcripts relative to that of a male-specific gene, the dynein heavy chain gene (PBANKA_0416100). The result was comparable to when *p28* was used as a control; the relative transcript level of *psh3* was more than 3-fold higher in motif_mutated than in motif_WT (Fig 5C, right graph). Collectively, these results indicated that the major motif functions as a *cis*-regulatory element for repressing downstream genes.

## *Pb*AP2-FG2 requires a co-repressor *Pb*AP2R-2 to repress its target genes

We previously identified two putative transcriptional regulator genes, *pbap2r-1* (PBANKA_0612400) and *pbap2r-2* (PBANKA_1418100), as a target gene of *Pb*AP2-G and *Pb*AP2-FG [20]. Of these, *pbap2r-1* functions as a transcriptional activator in zygotes and is renamed *pbap2-z* [25], but the functional role of *pbap2r-2* remains unknown. *Pb*AP2R-2 has an ACDC domain at its C-terminus. A previous study indicated that *Pf*AP2-2 has two putative AP2 domain [42], one of which is not highly conserved in the *Plasmodium* species (S3 Fig). The other one had AP2-like structure in both *P. berghei* and *P. falciparum*, but only the third beta-sheet and the alpha-helix were highly conserved between the two species. Because three-stranded antiparallel beta-sheets in AP2 are the structure that interacts with DNA [43,44], it was considered that this AP2-like domain is probably not involved in binding to DNA of a specific sequence. *Pb*AP2R-2 is expressed in females, and *pbap2r-2* knockout parasites [*pbap2r-2*(-)] are not able to form banana-shaped ookinetes [20]. Furthermore, in the recent study by Russell *et al.*, *pbap2r-2* was identified as one of the genes critical for female differentiation (referred as *fd3*) [45]. Therefore, we hypothesized that *Pb*AP2R-2 might play an essential role in female development and assessed its function. First, we performed a cross-fertilization assay to confirm the role of *Pb*AP2R-2 in female gametocyte development. When crossed with *p47*(-), *pbap2r-2*(-) could not produce normal ookinetes, confirming that development of female gametocytes in *pbap2r-2*(-) was impaired (Fig 6A). On the other hand, when crossed with *p48/45*(-), *pbap2r-2*(-) produced mature ookinetes nearly as many as when *p47*(-) and *p48/45*(-) were crossed (Fig 6A). These results revealed that as suggested by the previous studies, *Pb*AP2R-2 is essential for female development. Next, we developed a parasite line expressing GFP-fused *Pb*AP2R-2 using CRISPR/Cas9 (*Pb*AP2R-2::GFP^C, S2D Fig). The female gametocytes of *Pb*AP2R-2::GFP^C showed nuclear-localized fluorescence, consistent with our previous study; therefore, we performed ChIP-seq analysis at the gametocyte stage. Through this analysis, we obtained 944 highly reproducible peaks (IDR < 0.01) from 1597 and 1459 peaks identified in Experiments 1 and 2, respectively (S4A and S4B Table). Intriguingly, the genome-wide peak patterns for ChIP-seq of *Pb*AP2-FG2 and *Pb*AP2R-2 seemed highly similar

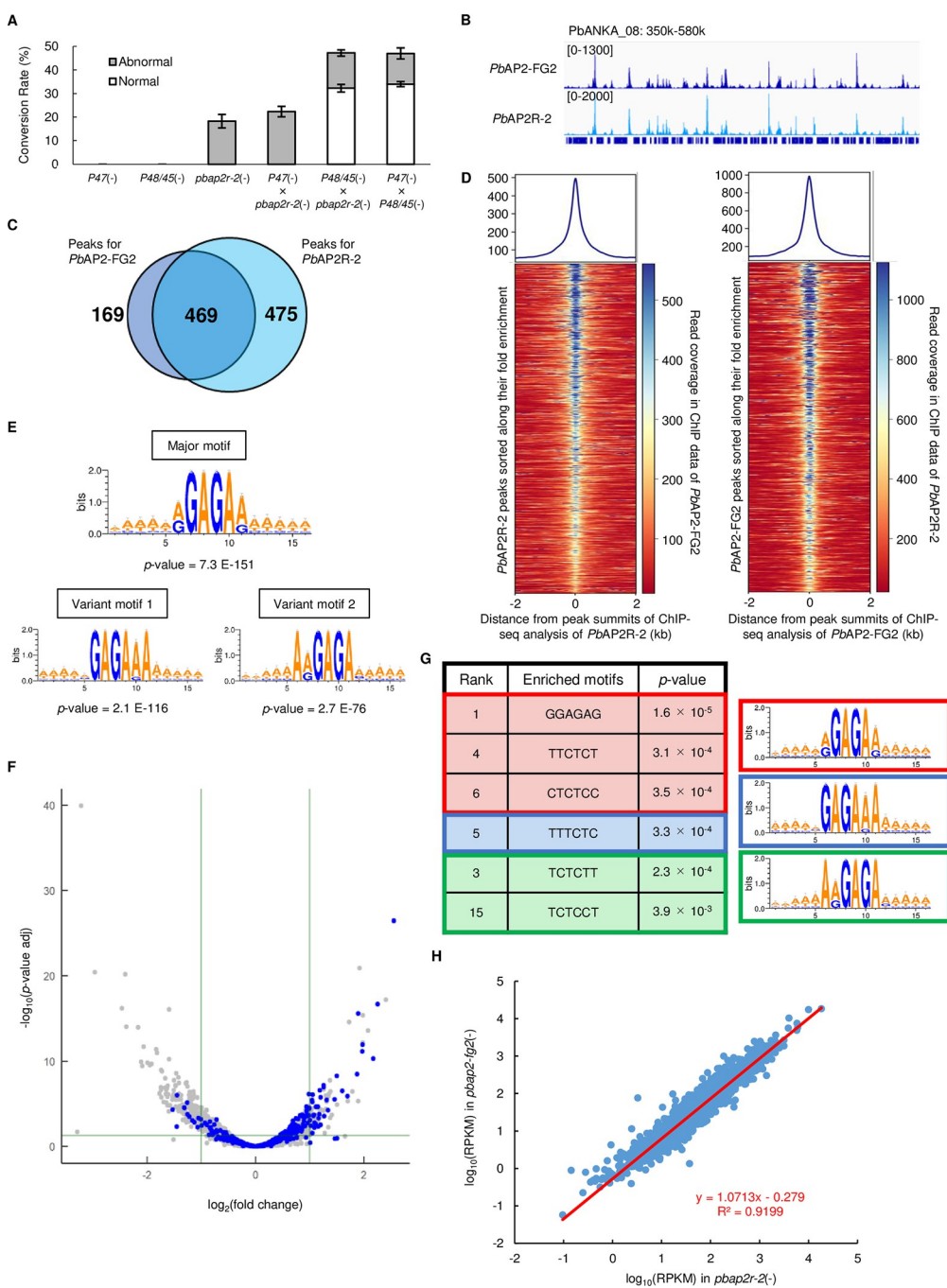

**Fig 6. ChIP-seq of *Pb*AP2R-2 and differential expression analysis between WT and *pbap2r-2*(-).** (A) Cross-fertilization assay among *pbap2r-2*(-), *p48/45*(-) and *p47*(-). The number of normal and abnormal ookinetes are indicated as white and grey bars, respectively. Error bars indicate the standard error of the mean (n = 3). (B) IGV images showing peaks identified in the ChIP-seq analysis of *Pb*AP2-FG2 and *Pb*AP2R-2. Read coverage for the ChIP data is shown. (C) A Venn diagram showing the overlap between peaks detected in the ChIP-seq of *Pb*AP2-FG2 and *Pb*AP2R-2. (D) Heat maps showing coverage in ChIP-seq of *Pb*AP2-FG2 at *Pb*AP2R-2 peaks (left) and coverage in ChIP-seq of *Pb*AP2R-2 at *Pb*AP2-FG2 peaks (right). Peak regions are aligned in ascending order of their fold enrichment value. Graphs on top of the heat maps show the mean coverage of all peak regions. (E) Motifs enriched within 50 bp from peak summits identified in the ChIP-seq of *Pb*AP2R-2. The logos were depicted using WebLogo 3. (F) A volcano plot showing DEGs in *pbap2r-2*(-) compared to WT. Blue dots represent the target genes of *Pb*AP2-FG2. A horizontal line indicates a *p*-value of 0.05, and two vertical lines indicate log$_2$(Fold Change) of -1 and 1. (G) Six-bp DNA motifs enriched within upstream region of genes upregulated in *pbap2r-2*(-). (H) A scatter plot showing relationship between log$_{10}$(RPKM) in *pbap2-fg2*(-) and *pbap2r-2*(-) for each gene. A red line indicates a linear approximation of the plots.

(Fig 6B), and more than 70% of the *Pb*AP2-FG2 peaks were overlapped with the *Pb*AP2R-2 peaks (Fig 6C). To evaluate the consistency of the ChIP peak pattern over the whole genome, we assessed the read coverage in the ChIP-seq data of *Pb*AP2-FG2 at the peak summits identified in the ChIP-seq of *Pb*AP2R-2 and *vice versa*. The results depicted that read coverage for *Pb*AP2-FG2 was enriched at the *Pb*AP2R-2 peaks, and higher fold enrichment of *Pb*AP2R-2 peaks correlated with higher read count detected in the *Pb*AP2-FG2 ChIP-seq (Figs 6D S4). The same was true for the read coverage of the *Pb*AP2R-2 ChIP-seq at the *Pb*AP2-FG2 peaks, indicating that the peak patterns in the ChIP-seq of *Pb*AP2-FG2 and *Pb*AP2R-2 were genome-widely consistent. Consistently, motif enrichment analysis by Fisher's exact test revealed that the major and two variant motifs were all enriched within 100 bp from the summits of peaks identified in the ChIP-seq of *Pb*AP2R-2 ($p$-value = $7.3 \times 10^{-151}$, $2.1 \times 10^{-116}$ and $2.7 \times 10^{-76}$ for the major motif, variant motifs 1 and 2, respectively, Fig 6E). Collectively, these results suggested that *Pb*AP2R-2 colocalizes with *Pb*AP2-FG2 on the genome and may cooperatively work with *Pb*AP2-FG2 during female development.

To evaluate the functions of *Pb*AP2R-2 in the expression of its targets, we performed differential expression analysis by comparing WT and *pbap2r-2*(-). We identified 95 significantly upregulated and 222 significantly downregulated genes (Fig 6F, S5A, S5B, and S5C Table). Among the upregulated genes, 36 targets of *Pb*AP2-FG2 were detected, and overall, the target genes tended to be upregulated in *pbap2r-2*(-) with a $p$-value of $1.6 \times 10^{-53}$ by two-tailed Student's t-test. This upregulation tendency was also detected within female- and male-enriched genes with a $p$-value of $5.6 \times 10^{-5}$ and $4.2 \times 10^{-8}$, respectively. Moreover, in the upstream region of the genes upregulated in *pbap2r-2*(-), one of the major motifs, GGAGAG, was most enriched with a $p$-value of $1.6 \times 10^{-5}$ by Fisher's exact test, and some other binding motifs of *Pb*AP2-FG2/*Pb*AP2R-2 were also significantly enriched (Fig 6G). Therefore, we considered that *Pb*AP2R-2 functions as a transcriptional repressive factor in female gametocytes. The upregulated and downregulated genes identified in *pbap2-fg2*(-) and *pbap2r-2*(-) presented a significant overlap with a $p$-value of $3.4 \times 10^{-37}$ and $4.1 \times 10^{-62}$ by Fisher's exact test, respectively. To further evaluate the relationship between *pbap2-fg2*(-) and *pbap2r-2*(-) in detail, we plotted RPKM value in *pbap2-fg2*(-) against that in *pbap2r-2*(-) for each gene. We then constructed a linear approximation of the plots, which depicted a line with a slope of 1.07 and a mean square correlation coefficient of 0.92 (Fig 6H). This result indicated that the change in the expression level for each gene was mostly comparable between *pbap2-fg2*(-) and *pbap2r-2*(-); hence, disruption of *pbap2-fg2* and *pbap2r-2* had highly similar effects on the female transcriptome. Therefore, we hypothesized that *Pb*AP2R-2 may function as an essential co-repressor of *Pb*AP2-FG2 in female gametocytes.

## *Pb*AP2-FG2 and *Pb*AP2R-2 form a transcriptional repressor complex

To investigate whether *Pb*AP2-FG2 and *Pb*AP2R-2 function as a transcriptional repressor complex on the genome, we performed rapid immunoprecipitation mass spectrometry of endogenous proteins (RIME) [46,47]. RIME is a method combining ChIP and MS to identify proteins that form a complex with a target protein of ChIP. We performed ChIP experiments with *Pb*AP2-FG2::GFP and WT, as a negative control, by the same method as ChIP-seq analysis and then on-beads digested the immunoprecipitated proteins with trypsin. The released peptides were then analyzed by liquid chromatography-tandem MS (LC-MS/MS) analysis. Four biologically independent experiments were performed for each parasite line, and immunoprecipitated proteins were identified. The analysis detected 482 and 463 proteins for *Pb*AP2-FG2::GFP and WT, respectively (Fig 7A and S6 Table). As a possible interaction partner of *Pb*AP2-FG2, proteins that were unique or more than fivefold enriched in *Pb*AP2-FG2

**A**

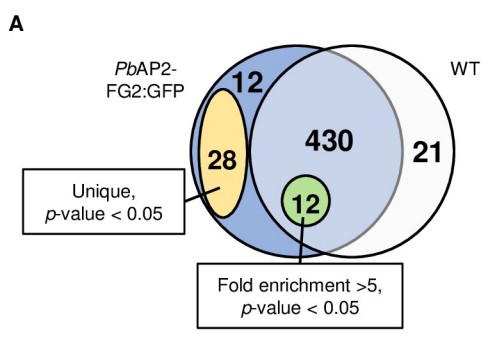

**B**

| ID | annotation | average QV | log₂(fold enrichment) | *p*-value |
|---|---|---|---|---|
| PBANKA_1015500 | *Pb*AP2-FG2 | 29.9 | (unique) | 4.46E-5 |
| PBANKA_1418100 | *Pb*AP2R-2 | 22.1 | (unique) | 1.02E-3 |
| PBANKA_1231600 | *Pb*AP2-O2 | 18.0 | (unique) | 1.16E-3 |
| PBANKA_1034300 | *Pb*AP2-G2 | 6.6 | (unique) | 5.60E-3 |
| PBANKA_1331400 | *Pb*MORC, putative | 37.8 | 3.64 | 1.32E-3 |
| PBANKA_1009500 | glutamate synthase [NADH], putative | 12.2 | 2.59 | 5.28E-4 |
| PBANKA_1123500 | SNF2 helicase, putative | 9.4 | 3.74 | 1.01E-3 |
| PBANKA_1241800 | DNA replication licensing factor MCM3, putative | 7.3 | 3.22 | 1.04E-3 |
| PBANKA_0616700 | NIMA related kinase 4 | 5.3 | 4.58 | 1.45E-5 |

**Fig 7. RIME using *Pb*AP2-FG2::GFP.** (A) A Venn diagram showing the overlap between proteins detected in the RIME using *Pb*A2-FG2 and WT. Number of proteins that were unique and more than fivefold enriched with *p*-value < 0.05 in *Pb*AP2-FG2::GFP compared to WT was indicated in a yellow and green circle, respectively. (B) A list of possible interaction partners of *Pb*AP2-FG2 identified in the RIME. Those with average quantitative value > 5 for *Pb*AP2-FG2::GFP (average QV) are shown.

compared to WT were searched according to the criteria used in the previous RIME study [47], and 40 unique and 12 fivefold enriched proteins were identified (Fig 7A). Of the proteins that were unique for *Pb*AP2-FG2::GFP, *Pb*AP2-FG2 was detected with the highest quantitative value as expected (Fig 7B). The unique protein with the second highest value was *Pb*AP2R-2. This result revealed that *Pb*AP2-FG2 and *Pb*AP2R-2 actually form a complex on the genome. The third unique protein was *Pb*AP2-O2 (encoded by PBANKA_1231600), an AP2 transcription factor essential for ookinete development [28]. A study in *P. yoelii* reported that *Py*AP2-O2 was expressed at several stages including gametocyte [29]. Therefore, *Pb*AP2-O2 might function as a transcriptional repressor in gametocyte together with *Pb*AP2-FG2.

In the proteins identified as an interaction partner of *Pb*AP2-FG2, we also found *Pb*MORC (encoded by PBANKA_1331400) with average quantitative value of 37.8, which was the highest in the fivefold enriched proteins, and log₂(fold enrichment) of 3.64 (Fig 7B). Microrchidia (MORC) is a nuclear protein that contains a GHKL (gyrase, Hsp90, histidine kinase, and MutL)-ATPase domain [48]. The MORC family proteins in animals and plants are involved in chromatin condensation and remodeling [49,50]. In an apicomplexan parasite *Toxoplasma gondii*, MORC interacts with AP2 transcription factors and plays a role in transcriptional repression of genes downstream of its binding sites [51]. Therefore, *Pb*MORC might be recruited by *Pb*AP2-FG2 and play a role in the transcriptional repression through remodeling of chromatins.

## *Pb*AP2-FG2 and *Pb*AP2R-2 repress the target genes of AP2-G

In previous studies, we observed the expression patterns of *Pb*AP2-G and *Pb*AP2-FG and found that the expression of *Pb*AP2-FG begins during the period when the expression of *Pb*AP2-G decreases [20,23]. This result indicates that a major transcriptional activator switches from *Pb*AP2-G to *Pb*AP2-FG as early gametocytes develop into females. Considering this scenario, we hypothesized that the *Pb*AP2-FG2-*Pb*AP2R-2 complex might support stage conversion from early gametocytes to female gametocytes by repressing early gametocyte genes activated by *Pb*AP2-G. To address this possibility, we assessed whether the target genes of *Pb*AP2-FG2 include those of *Pb*AP2-G. Consistent with our hypothesis, we found a significant overlap (105 genes) between the target genes of *Pb*AP2-G and *Pb*AP2-FG2 with a *p*-value of $4.5 \times 10^{-7}$ by Fisher's exact test (Fig 8A). In addition, the distribution of log₂(fold change) values for these common target genes in the differential expression analysis between WT and

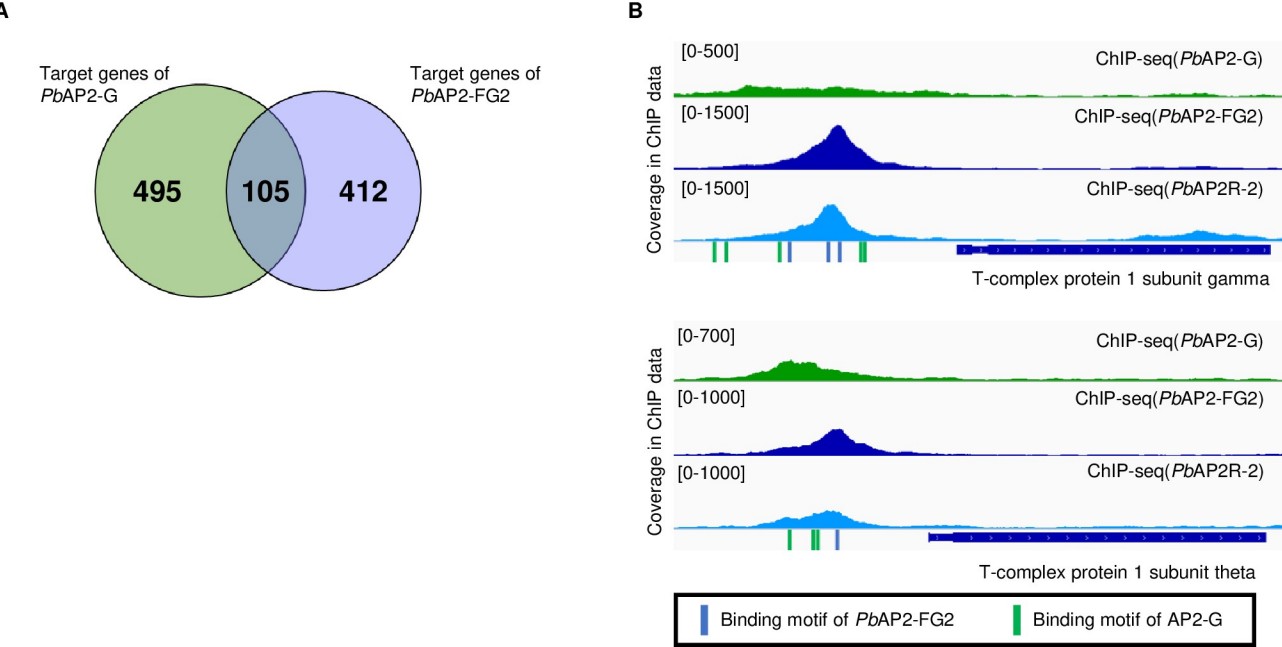

**Fig 8. Relationship between target genes of *Pb*AP2-G and *Pb*AP2-FG2.** (A) A Venn diagram showing the overlap between target genes of *Pb*AP2-G and *Pb*AP2-FG2. (B) IGV images showing representative peaks of ChIP-seq for *Pb*AP2-G, *Pb*AP2-FG2, and *Pb*AP2R-2 on the upstream of target genes common for *Pb*AP2-G and *Pb*AP2-FG2. Read coverage for the ChIP data is shown. Positions of the *Pb*AP2-G and *Pb*AP2-FG2 binding motifs are indicated in green and blue, respectively.

*pbap2-fg2*(-) was higher than that for the non-target genes with *p*-value = $1.4 \times 10^{-13}$ by two-tailed Student's t-test, suggesting that these genes are actually repressed in female gametocytes.

Most of these common target genes between *Pb*AP2-FG2 and *Pb*AP2-G have not been assessed for their function during *Plasmodium* gametocyte development. In the common target genes, we found T-complex protein 1 subunit (TCP-1) genes (Fig 8B, S3C Table). *Plasmodium* parasites possess eight TCP-1 genes, all of which are a target of *Pb*AP2-G. Although the target genes of *Pb*AP2-FG2 contained only four of these genes, peaks with fold enrichment > 2.5 were found upstream of three other TCP-1 genes in the ChIP-seq of *Pb*AP2-FG2. TCP-1s comprise a type 2 chaperonin, tailless complex polypeptide 1 ring complex (TRiC), which has been indicated to play essential roles in folding diverse polypeptides, including actin and tubulin [52–55]. In *P. falciparum*, it has been reported that TRiC is essential for the asexual blood-stage development [56,57]. Thus, this complex is presumed to be widely required in the *Plasmodium* life cycle, which, for *P. berghei*, includes early gametocyte development, but not during female development. Notably, the common target genes of *Pb*AP2-FG2 and *Pb*AP2-G also included the actin I gene, alpha-tubulin 2 gene, and a putative tubulin beta chain gene (S3C Table). These findings suggested the possibility that for female gametocytes of *P. berghei*, cytoskeletal development is mostly completed during early gametocyte development.

## The function of *Py*AP2-O3 is highly similar to that of *Pb*AP2-FG2

The present study revealed that transcriptional repression by the *Pb*AP2-FG2-*Pb*AP2R-2 complex is essential for regulating the female transcriptome. Recently, Li *et al.* reported that *Py*AP2-O3 is also a transcriptional repressor in female gametocytes, but their other conclusions differed from what we have revealed here [30]. The first is regarding the role of

*Py*AP2-O3, which was primarily based on their RNA-seq data. The authors performed RNA-seq analyses with female gametocytes collected by cell sorting using a female-specific fluorescent marker and compared the female transcriptome of *P. yoelii* 17XNL (*Py*WT) and *pyap2-o3*-null parasites. The analysis detected significant upregulation of 1141 genes in *pyap2-o3*-null parasites, more than half of which were specifically or preferentially expressed in males. Accordingly, the authors concluded that *Py*AP2-O3 globally represses male genes to safeguard the female transcriptome. This statement differed from our conclusion that *Pb*AP2-FG2 and *Pb*AP2R-2 repress various genes, which include some early gametocyte and female genes. The second is the conclusions derived from their ChIP-seq analysis of *Py*AP2-O3. For example, the binding motif of *Py*AP2-O3 predicted in their study was considerably different from that of *Pb*AP2-FG2 identified in this study; the binding motif of *Py*AP2-O3 was predicted to be TRTRTGCA. As *P. berghei* and *P. yoelii* are phylogenetically very close, such discrepancies in the roles of orthologous genes seems unlikely. Therefore, we reassessed the ChIP-seq and RNA-seq data deposited from the *Py*AP2-O3 study to clarify the inconsistency between the two studies.

First, we reanalyzed the ChIP-seq data for *Py*AP2-O3. We mapped their sequence data onto the *P. yoelii* reference genome (downloaded from PlasmoDB) using bowtie2 and removed reads aligned onto more than two sites of the genome, as in our ChIP-seq analysis. Then, we called peaks using macs2 with the criteria used by Li *et al.* (fold enrichment $> 2.0$ and $p$-value $< 1.0 \times 10^{-5}$) and obtained 1309 peaks that were common in the duplicates (S7A and S7B Table). Within 100 bp from these peak summits, we found enrichment of TRTRTGCA with a $p$-value of $7.0 \times 10^{-52}$ by Fisher's exact test (Fig 9A). On the other hand, the major motif identified in our study, RGAGAR, was also enriched with a $p$-value of $2.2 \times 10^{-52}$, comparable to that of the TRTRTGCA motif (Fig 9A). In addition, the variant motifs 1 and 2 were also enriched with $p$-values of $4.0 \times 10^{-48}$ and $9.4 \times 10^{-26}$, respectively. We further performed IDR1D analysis and found that only 122 peaks had an IDR $< 0.01$, which implies the low reproducibility of the ChIP-seq data (Fig 9B, 9C, S7A and S7B Table). Within these highly reliable peaks (IDR $< 0.01$), the RGAGAR motif was much more enriched ($p$-value $= 2.1 \times 10^{-12}$) than the TRTRTGCA motif ($p$-value $= 1.1 \times 10^{-4}$, Fig 9A). Therefore, in contrast to the major motif, the TRTRTGCA motif appeared to have been mainly derived from unreliable peaks (IDR $\geq 0.01$).

We further predicted the target genes of *Py*AP2-O3 from the 1309 peaks common in duplicates. The analysis identified 781 target genes, 271 of which were included in the target genes of *Pb*AP2-FG2 as an orthologous gene (Fig 9D and S7C Table). Although there was a significant overlap between the two target sets ($p$-value $= 2.4 \times 10^{-85}$ by Fisher's exact test), approximately 70% of the *Py*AP2-O3 targets were not included in the *Pb*AP2-FG2 targets. Given the low reproducibility of the ChIP-seq data, we supposed that some targets might have been derived from unreliable peaks and hence not a true target. We further assessed the sex-enriched expression of the target genes of *Py*AP2-O3 considering orthologous genes of the female- and male-enriched genes defined above as sex-enriched genes for *P. yoelii*. We found that the target genes contained only 94 male-enriched genes, which contradicted the conclusion that *Py*AP2-O3 globally represses male genes (Fig 9E). Moreover, the targets also contained 128 female-enriched genes, 58 orthologous genes of which were included in the targets of *Pb*AP2-FG2, implying that, similar to *Pb*AP2-FG2, *Py*AP2-O3 also plays a role in repressing a substantial number of female genes.

Next, we reassessed the RNA-seq data of the *Py*WT and *pyap2-o3*-null parasites. In their RNA-seq analysis, no threshold of fragments per kilobase of transcript per million mapped reads (FPKM) was set to exclude genes with low expression levels. Such an analytical process may detect DEGs derived from artificial variances and yield large DEG lists with large

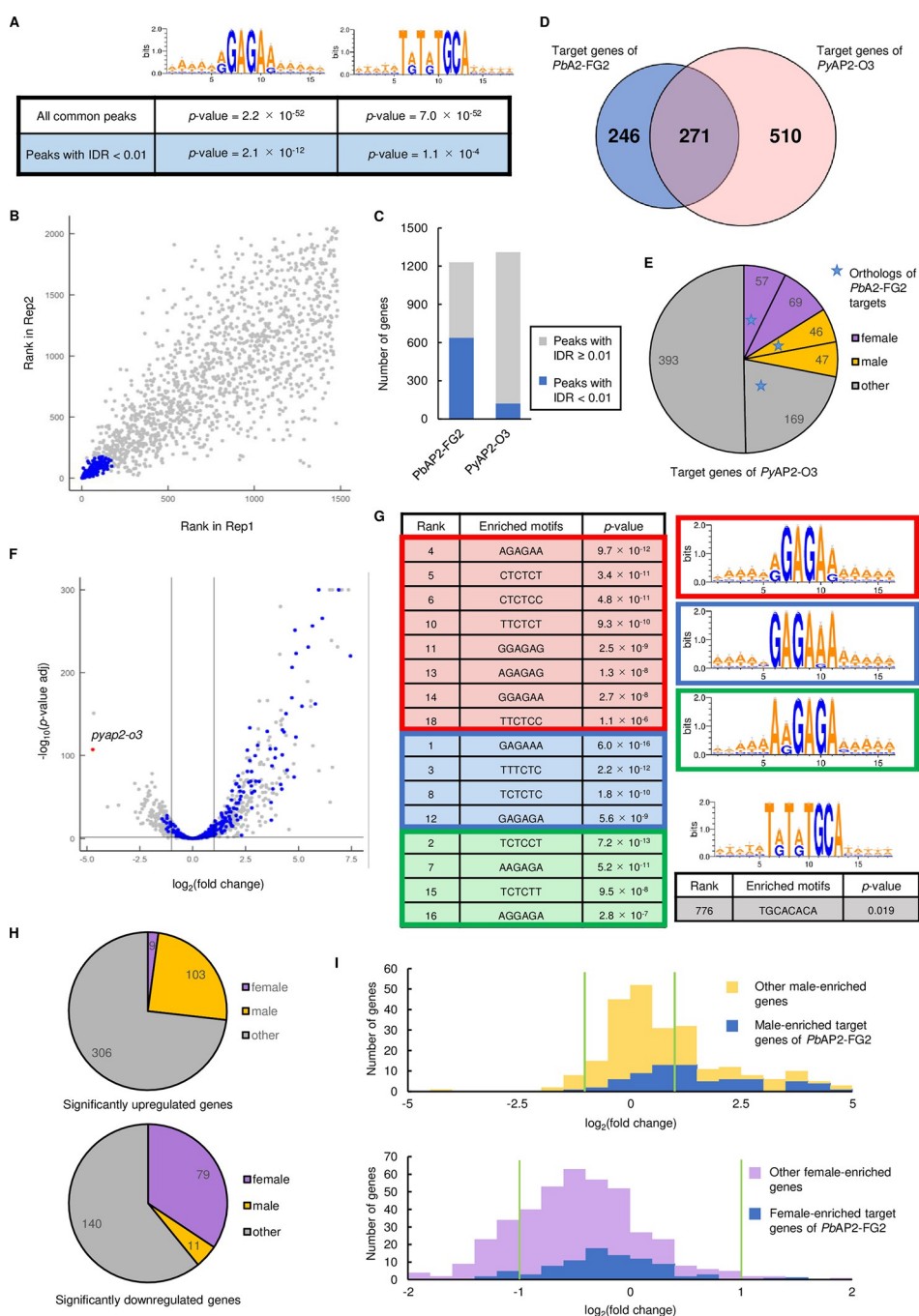

**Fig 9. Reassessment of the ChIP-seq and RNA-seq data for the study of *Py*AP2-O3.** (A) Enrichment of RGAGAR and TRTRTGCA motif in the peak region identified by ChIP-seq analysis of *Py*AP2-O3. The logos were depicted by WebLogo 3. (B) IDR1D analysis between the ChIP-seq experiment 1 and 2 of *Py*AP2-O3. Peaks with IDR < 0.01 are indicated as blue dots. (C) Ratio of peaks with IDR < 0.01 in all peaks identified in ChIP-seq of *Pb*AP2-FG2 and *Py*AP2-O3. (D) A Venn diagram showing number of genes common in the targets of *Pb*AP2-FG2 and *Py*AP2-O3. (E) Classification of the target genes of *Py*AP2-O3 into sexual stage-enriched gene sets. The number of target genes common for *Py*AP2-O3 and *Pb*AP2-FG2 are indicated with a blue star for each set. (F) A volcano plot showing DEGs in *pyap2-o3*-null parasite compared to *Py*WT. Blue dots represent orthologs of the target genes of *Pb*AP2-FG2, and a red dot indicates *pyap2-o3*. A horizontal line indicates a *p*-value of 0.05, and two vertical lines indicate log$_2$(Fold Change) of -1 and 1. (G) Six-bp DNA motifs enriched within the upstream region (300 to 1200 bp from ATG) of genes upregulated in *pyap2-o3*-null parasite. (H) Classification of significantly upregulated and downregulated genes into sexual stage-enriched gene sets (top and bottom, respectively). (I) Histograms showing distribution of log$_2$(Fold Change) values in *pyap2-o3*-null parasite for female and male-enriched genes (top and bottom, respectively). Green lines indicate log$_2$(Fold Change) of -1 and 1.

variance. In the analysis performed by Li *et al.*, more than one-third of the upregulated genes (445/1,141 genes) had FPKM < 10 for *pyap2-o3*-null parasites. Although these genes with low scores could demonstrate high-fold enrichment, the actual upregulation was low and could be false positives; thus, it is not appropriate to conclude the function of *Py*AP2-O3 from such analyses. To obtain more robust results, we analyzed the RNA-seq data according to the analytical process performed in this study, setting a minimum FPKM threshold of $\geq 10$ (S8A Table). This analysis identified 418 significantly upregulated and 230 significantly downregulated genes (Fig 9F, S8B and S8C Table). In the upregulated genes, the target genes of *Py*AP2-O3, which we obtained above, were enriched with a *p*-value of $7.7 \times 10^{-9}$ by Fisher's exact test. On the other hand, the orthologous genes of *Pb*AP2-FG2 targets were more enriched (*p*-value = $3.4 \times 10^{-17}$) (Fig 9F), again suggesting that the target list of *Py*AP2-O3 may contain several pseudo-targets. We next investigated whether the binding motifs of AP2-FG2 identified in this study and that by Li *et al.* were enriched in the upstream of these upregulated genes. Consistent with our results, the major motif and two variant motifs were highly enriched in the upstream region compared to that of the other genes (Fig 9G). In fact, all 6-bp motifs that belonged to the major motif, variant motif 1 or 2 were detected in the 20 most enriched motifs, suggesting that these motifs functioned as a *cis*-acting repressive element in *P. yoelii* as well. However, when enrichment of any 8-bp motif was searched in the same region, the TRTRTGCA motif was not found to be significantly enriched; that is, the most enriched motif that corresponds to TRTRTGCA was detected as the 776th enriched motif with a *p*-value of 0.019 (Fig 9G), indicating that this motif is not related to the upregulation of genes detected in *pyap2-o3*-null parasites.

According to our analysis, the majority of male-enriched genes were not significantly upregulated in *pyap2-o3*-null parasites; the upregulated genes only contained 103 male-enriched genes (Fig 9H and 9I). Moreover, nearly half of the upregulated male-enriched genes were not a target gene of *Pb*AP2-FG2 or *Py*AP2-O3. Therefore, based on our analysis of their RNA-seq data, it seemed not appropriate to conclude that *Py*AP2-O3 globally represses male genes. For female-enriched genes, only nine genes were significantly upregulated, and most genes tended to be downregulated, as discussed by Li *et al.* (Fig 9H and 9I). The downregulation of female genes was considered to be caused by impairment of the female transcriptome upon disruption of *pyap2-o3*. Therefore, we hypothesized that such an effect caused by the disruption of *pyap2-o3* might have masked the upregulation of female-enriched target genes in the RNA-seq analysis. In fact, despite the overall downregulation of female-enriched genes, most of the female-enriched target genes were not downregulated; there was a significant difference in the distribution of $\log_2$(fold change) between the female-enriched target genes and the other female-enriched genes (*p*-value = $1.7 \times 10^{-4}$ by two-tailed Student's t-test) (Fig 9I). These results suggested that consistent with the results for *Pb*AP2-FG2, *Py*AP2-O3 also targets female genes and represses their expression. Collectively, our analysis revealed that *Pb*AP2-FG2 and *Py*AP2-O3 both repress not only male genes but also a wide-variety of genes to support female differentiation.

## Discussion

This study highlights how *Pb*AP2-FG2 and *Pb*AP2R-2 function cooperatively as a transcriptional repressor complex during female development. Their target genes contained variable genes regarding functional annotation and expression patterns, which indicated that this repressor complex might play distinct roles for each group of the target genes during female development. As suggested by the comparison between the target genes of *Pb*AP2-FG2 and *Pb*AP2-G, one of the roles of the repressor complex could be repression of the early

gametocyte genes to promote female differentiation. During early gametocyte development, the female transcriptional activator *Pb*AP2-FG begins to be expressed as *Pb*AP2-G expression decreases [20,23]. Thus, we considered that the repression of *Pb*AP2-G targets was vital for completing this switch of major transcriptional activators. On the other hand, the target genes of *Pb*AP2-FG2 also included a significant number of female-enriched genes. This result appeared unreasonable because it would mean that such genes are activated and repressed during the same period. Nevertheless, the differential expression analysis suggested that these female-enriched target genes were indeed repressed by AP2-FG2/AP2-O3; for both *P. berghei* and *P. yoelii*, expression of the female-enriched target genes was not downregulated in *ap2-fg2/ap2-o3*-knockout parasites despite that the other female-enriched genes were predominantly downregulated. This observation implied that transcriptional activators alone could not precisely control gene expression for female gametocyte development, thereby requiring repressors for its modulation. Another possible role of the *Pb*AP2-FG2-*Pb*AP2R-2 repressor complex was the repression of male genes, as suggested in the study of *Py*AP2-O3 [30]. However, the results in this study did not corroborate their conclusion that the global repression of male genes by *Py*AP2-O3 was required for balancing the female-specific transcriptome. The ChIP-seq analyses of AP2-FG2/AP2-O3 in *P. berghei* and *P. yoelii* both showed that the target genes of AP2-FG2/AP2-O3 contained only a subset of male-enriched genes. These included only a limited number of genes related to the major characteristic features of male gametocytes/microgametes, such as flagella formation and DNA replication. Moreover, although a significant number of male-enriched genes were upregulated in *pyap2-o3*-null parasites, nearly half of them were not a target gene of *Py*AP2-O3. From this result, we speculated that the target genes of AP2-FG2/AP2-O3 could contain male-specific transcriptional regulator genes, especially activators; the non-target genes were indirectly upregulated in *pyap2-o3*-null parasites, as the expression of male transcriptional activators was released from the repression by *Py*AP2-O3. Therefore, we proposed that the repressor complex repressed a subset of male genes that include those essential for regulating male differentiation. Very recently, Russel *et al.* identified five genes that are essential for male differentiation [45], but these genes were not included in the target gene list of *Pb*AP2-FG2 and *Pb*AP2R-2. Collectively, we concluded that the *Pb*AP2-FG2-*Pb*AP2R-2 complex could play three roles: promoting female differentiation by repressing early gametocyte genes, modulating the expression level of female genes, and suppressing male differentiation.

In this study, we performed RIME on the *Pb*AP2-FG2::GFP parasite and identified *Pb*MORC as one of the proteins that form a complex with *Pb*AP2-FG2. Previous studies in apicomplexan parasites reported that MORC interacts with diverse AP2 transcription factors [22,51,58]. In *P. falciparum*, *Pf*MORC was found to interact with an AP2 transcriptional repressor, *Pf*AP2-G2 [22]. In addition, a study in *Toxoplasma gondii* reported that MORC forms a complex with several AP2 transcription factors and plays a role in transcriptional repression [51]. These studies were consistent with our result that *Pb*MORC is an interaction partner of a transcriptional repressor *Pb*AP2-FG2 and hence indicate possibility that in Apicomplexa, transcriptional repression occurs in a similar mechanism involving MORC and AP2 transcription factors. The study in *Toxoplasma* reported that MORC is also involved in recruitment of histone deacetylase 3, which plays a role in gene repression through reversing histone acetylation, a mark for transcriptional activation [59,60]. However, any histone deacetylase was not found as an interaction partner of *Pb*AP2-FG2 in our RIME or that of *Pf*AP2-G2 by IP-MS [22]. Given that MORC has a conserved ATPase domain and is known to play a role in chromatin regulation in animals and plants, a major function of MORC in *Plasmodium* might be related to ATPase activity and chromatin regulation but not recruitment of other factors, such as histone deacetylase.

Our result demonstrated that the RIME method is a powerful tool for identifying proteins in a transcriptional complex in *Plasmodium*. By cross-linking complexes in their native state using formaldehyde, transient protein-protein interactions can be preserved in the RIME [46]. Furthermore, the formaldehyde fixation allows stringent washing regimes, which reduce background nonspecific noise in the MS analysis [46]. Although recent studies in AP2 and other sequence specific transcription factors have advanced our knowledge on stage specific transcriptional regulation of *Plasmodium*, the mechanisms of transcriptional activation/repression by the transcription factors and their cofactors have been poorly investigated. We believe that further use of the RIME method on *Plasmodium* transcription factors would identify components of transcriptional complexes and considerably help us understand detailed mechanisms of transcriptional regulation in the parasites.

In *P. berghei*, another AP2 transcriptional repressor gene, *pbap2-g2*, is activated by *Pb*AP2-G [20], a transcriptional activator that triggers sexual differentiation, similar to *pbap2-fg2*, which is activated by the female transcriptional activator *Pb*AP2-FG. *Pb*AP2-G2 induces global gene repression, and disruption of this gene results in the aberrant development of gametocytes. The target genes of *Pb*AP2-G2 include various genes, like those of *Pb*AP2-FG2 do, suggesting that *Pb*AP2-G2 also plays multiple roles, possibly repressing trophozoite genes, modulating the expression of early gametocyte genes, and suppressing asexual fate. Studies of *Plasmodium* AP2-family proteins have shown that transcriptional regulation by AP2 transcription factors is remarkably simple; one transcription factor comprehensively activates certain stage-specific genes [20,23,25,61,62]. On the other hand, recent studies of repressors have indicated that to regulate stage conversion, such as sexual differentiation and sex determination, transcriptional repressors also play an essential role [21,22,30,63]. In addition, studies in another apicomplexan parasite, *Toxoplasma*, have reported an essential role for transcriptional repressors in conversion from tachyzoite to bradyzoite [64,65]. Therefore, it was suggested that optimal transcriptional regulation could not be controlled by transcriptional activators alone during stage conversion in Apicomplexa. This observation suggested the possibility of finding transcriptional repressors during the other stage conversion processes as well, and investigating these factors would help us understand the mechanisms promoting the life cycle of *Plasmodium*.

## Materials and methods

### Ethics statement

All experiments in this study were performed following the recommendations in the Guide for the Care and Use of Laboratory Animals of the National Institutes of Health to minimize animal suffering and were approved by the Animal Research Ethics Committee of Mie University, Mie, Japan (permit number 23–29).

### Parasite preparation

*pbap2-fg2*(-), *Pb*AP2-FG2::GFP, and *pbap2r-2*(-) were derived from the *P. berghei* ANKA strain. The other transgenic parasites were derived from Pbcas9 [39]. Parasites were inoculated in Balb/c or ddY mice. Ookinete cultures were performed at 20°C using RPMI1640 medium, pH 8.0, supplemented with fetal calf serum and penicillin/streptomycin at final concentrations of 20% and 100 U/mL, respectively. Cross-fertilization assays were conducted as previously described [66].

### Generation of mutant parasites

The DNA constructs for tagging *Pb*AP2-FG2 with GFP and knocking out *pbap2-fg2* were prepared as previously reported [66,67]. Briefly, for *gfp*-tagging, two homologous regions were

cloned into the *gfp*-fusion vector to fuse *pbap2-fg2* in-frame with *gfp*. The *gfp*-fusion vector was previously generated on the backbone of pBluescript SK (+) (Stratagene, La Jolla, CA, USA). In this vector, 3' untranslated region (UTR) of *hsp70* and a *hdhfr* expression cassette (under the control of *pbef-1α* promoter and 3' UTR of *pbdhfr*) were located 3'-side of the *gfp* gene in this order. Of the two homologous regions, one was cloned to 5'-side of the *gfp* using *Xho*I/*Nhe*I site, and the other was cloned to 3'-side of the *hdhfr* cassette using *Bam*HI/*Not*I site (S2A Fig). The plasmid was linearized by *Xho*I and *Not*I digestion before use in transfection experiments. To knock out *pbap2-fg2*, the targeting construct was prepared using overlap PCR. The construct had two homologous regions around the *pbap2-fg2* locus flanking a *hdhfr* expression cassette, which was derived from the *gfp*-fusion vector described above. *pbap2r-2*(-) was generated in a previous study [20].

The other transgenic parasites were generated by the CRISPR/Cas9 system using the parasites expressing Cas9 [39]. The Cas9-expressing parasite Pbcas9 has a *cas9* cassette at the *p230p* locus. The *hsp70* promoter controls the expression of Cas9, and Pbcas9 constitutively expresses Cas9 throughout the asexual blood cycle. Donor DNA for transfection was constructed by overlap PCR, cloned into pBluescript KS (+) using the *Xho*I and *Bam*HI sites by In-Fusion cloning, and then amplified by PCR from the constructed plasmid. sgRNA vectors were constructed as previously described [39]. Target sites of sgRNA were designed using the online tool CHOPCHOP (https://chopchop.cbu.uib.no/).

Transfection was performed using the Amaxa Basic Parasite Nucleofector Kit 2 (LONZA). All transfectants were selected by treating mice with 70 μg/mL pyrimethamine in their drinking water. Recombination was confirmed by PCR and, for the motif_mutated parasite, Sanger sequencing. The PCRs were performed using PrimeSTAR GXL DNA Polymerase (Takara). PCR products were amplified for 30 cycles, and annealing temperature was set at 55˚C. Clonal parasites were obtained by limiting dilution. All primers used in this study are listed in S9 Table (No. 1–48).

## Fluorescence analysis

Expression of GFP-fused *Pb*AP2-FG2 was analyzed by live fluorescence microscopy. Cells were stained with 1 ng/mL Hoechst 33342 for 10 min. Images were taken by Olympus BX51 microscope with Olympus DP74 camera.

## RNA-seq and sequence data analysis

The total RNA was extracted from parasites enriched with gametocytes by treating infected mice with 10 mg/L sulfadiazine in their drinking water for two days, using the Isogen II reagent (Nippon gene). Briefly, whole blood was withdrawn from infected mice and passed through the Plasmodipur filter to remove white blood cells, and the red blood cells (RBCs) were lysed in an ice-cold 1.5 M $NH_4Cl$ solution. After the lysis, the cells were subjected to Isogen II (NIPPON GENE), and the total RNA was extracted according to the manufacturer's instructions. RNA-seq libraries were prepared from the total RNA using the KAPA mRNA HyperPrep Kit (Kapa Biosystems) and sequenced using Illumina NextSeq. Three biologically independent experiments were conducted for each parasite line. The obtained sequence data were mapped onto the reference genome sequence version 3 of *P. berghei*, downloaded from PlasmoDB 46, using HISAT2, setting the parameter for maximum intron length to 1000. The mapping data for each sample were analyzed using featureCounts and compared using DESeq2. Genes in the subtelomeric regions were removed from the differential expression analysis. The parameters for all programs were set as the default unless otherwise specified.

For comparison between results for *P. berghei* in this study and *P. yoelii* in the previous study, 4426 orthologues were considered according to the orthology and synteny data in PlasmoDB.

## ChIP-seq and sequencing data analysis

Whole blood was withdrawn from the infected mice treated with sulfadiazine and passed through the Plasmodipur filter to remove white blood cells. The blood was diluted in a complete medium (RPMI1640 supplemented with 20% fetal calf serum) and immediately fixed with 1% formalin at 30°C. After fixing, RBCs were lysed in ice-cold 1.5 M NH4Cl solution. This step was performed several times until the supernatant became clear, and the cells were lysed in SDS lysis buffer (50 mM Tris-HCl, 1% SDS, 10 mM EDTA). The samples were sonicated at 4°C using Bioruptor (Cosmo Bio) for 20 cycles of 30 sec on/30 sec off to shear the chromatin. Input samples were collected at this point from the sonicated cell lysate. For IP samples, chromatins were immunoprecipitated with anti-GFP polyclonal antibodies (5 mg/mL; Abcam, ab290), which were bound to Protein A Magnetic Beads (Invitrogen) before the ChIP step (2 μL of antibody was mixed with 20 μL of beads and incubated for > 1 h). After 12 h of incubation at 4°C, the beads were washed with low-salt buffer (20mM Tris-HCl, 0.1% SDS, 1% Triton X-100, 2 mM EDTA, 150 mM NaCl) for five times and high-salt buffer (20mM Tris-HCl, 0.1% SDS, 1% Triton X-100, 2 mM EDTA, 500 mM NaCl) for three times. Immunoprecipitated chromatin was then eluted in elution buffer (10mM Tris-HCl, 1% SDS, 5 mM EDTA, 300 mM NaCl), heated at 65°C for 8 h, and processed with RNase H for 1 h and proteinase K for 2 h. DNA fragments were purified from the immunoprecipitated chromatin by phenol/chloroform extraction and ethanol precipitation and used for library construction. Libraries for NGS were prepared using the KAPA HyperPrep Kit (Kapa Biosystems) according to the manufacturer's instructions and sequenced using Illumina NextSeq. Two biologically independent experiments were performed for each sample and used for the following analysis.

The obtained sequence data were mapped onto the reference genome sequence version 3 of *P. berghei*, downloaded from PlasmoDB 46, using Bowtie 2. Reads aligned onto more than two sites were removed from the mapping data. Using the trimmed mapping data, peaks were called with macs2 callpeak with fold enrichment > 3.0 and *q*-value < 0.01. To identify reliable peaks, the data obtained from two biologically independent experiments were compared using IDR1D analysis (https://idr2d.mit.edu/) setting max gap to 100. Briefly, the peaks were ranked for each replicate according to their *p*-value. The peaks from each replicate were then compared and scored based on their respective ranks. Highly reproducible peaks were defined as those with an IDR score < 0.01. Binding motifs were predicted by analyzing the enrichment of motifs within 50 bp of peak summits using Fisher's exact test (the method was previously described in detail [61]). Genes with peaks within upstream of 1200 bp from ATG were identified as target genes. The parameters for all programs were set as the default unless otherwise specified.

## ChIP-qPCR and RT-qPCR for reporter experiments

ChIP experiments were performed as described for ChIP-seq analysis. IP and input samples were each prepared from 200 μL and 10 μL of sonicated cell lysate, respectively. Quantification of DNA fragments of interest was performed by real-time qPCR using TB Green Fast qPCR Mix (Takara) and Thermal Cycler Dice Real Time System II (Takara). Amplification cycles were performed for 40 cycles, and cycle threshold (Ct) was detected between 20 to 35 cycles. From the Ct values, %input values were calculated as $2\wedge(Ct^{IP}-Ct^{input}) \times (1/20) \times 100$. Three biologically independent experiments were performed and used for the analysis.

For RT-qPCR analyses, cDNA was synthesized from 1 μg of total RNA, extracted as described for RNA-seq analysis, using the PrimeScript RT reagent Kit with gDNA Eraser (Takara), and 1/200 of the synthesized cDNA was used for qPCR. Real-time qPCR experiments were performed as described above. All primers used are listed in S9 Table (No. 49–62).

## RIME

RIME was performed according to the previous studies [46,47] with some modifications. ChIP was performed as described for ChIP-seq analysis until washing beads with high salt buffer. After the wash, beads were washed twice with 100 mM ammonium hydrogen carbonate (AMBIC) solution. The bead-bound proteins were digested with 10 μl of trypsin (Promega) in 100 mM AMBIC at an enzyme-to-protein ratio of 1:100 (wt/wt) for overnight at 37˚C. After the overnight digest, additional 10 μl of trypsin was added, and the beads were further incubated for 4 h at 37˚C. The supernatant, which contains the digested peptides, was added to 100% formic acid, resulting in the final concentration of formic acid 5% (vol/vol). The digested peptides were purified with C18 tip (GL-Science, Tokyo, Japan) and then subjected to nanocapillary reversed-phase LC-MS/MS analysis using a C18 column (12 cm × 75 μm, 1.9 μm, Nikkyo technos, Tokyo, Japan) on a nanoLC system (Bruker Daltoniks, Bremen, Germany) connected to a timsTOF Pro mass spectrometer (Bruker Daltoniks) and a modified nano-electrospray ion source (CaptiveSpray; Bruker Daltoniks). The mobile phase consisted of water containing 0.1% formic acid (solvent A) and acetonitrile containing 0.1% formic acid (solvent B). Linear gradient elution was carried out from 2% to 35% solvent B for 20 min at a flow rate of 250 nL/min. The ion spray voltage was set at 1.6 kV in the positive ion mode. Ions were collected in the trapped ion mobility spectrometry (TIMS) device over 100 ms and MS and MS/MS data were acquired over an m/z range of 100–2,000. During the collection of MS/MS data, the TIMS cycle was adjusted to 0.53 s and included 1 MS plus 4 parallel accumulation serial fragmentation (PASEF)-MS/MS scans, each containing on average 12 MS/MS spectra (>100 Hz) [68,69], and nitrogen gas was used as collision gas.

## Data analysis for RIME

The MS/MS data obtained by RIME was processed using DataAnalysis version 5.2 (Bruker Daltoniks), and proteins were identified using MASCOT version 2.7.0 (Matrix Science, London, UK) against the Uniprot_Plasmodium_berghei_ANKA_strain database (4,948 sequences; 3,412,795 residues). Protease specificity was set for trypsin (C-term, KR; Restrict, P; Independent, no; Semispecific, no; two missed and/or nonspecific cleavages permitted). Variable modifications considered were N-terminal Gln to pyro-Glu, and oxidation of methionine. The mass tolerance for precursor ions was ±15 ppm. The mass tolerance for fragment ions was ±0.05 Da. The threshold score/expectation value for accepting individual spectra was $p < 0.05$. Quantitative value and fold enrichment were calculated by Scaffold5 version5.1.2 (Proteome Software, Portland, OR, USA) [70] and Microsoft Excel, respectively, for MS/MS-based proteomic studies. Proteins that were unique or more than fivefold enriched with $p$-value $< 0.05$ by two-tailed Student's t-test in *Pb*AP2-FG2 compared to WT were identified as a possible interaction partner of *Pb*AP2-FG2.

## Supporting information

**S1 Fig. An Integrative Genomics Viewer (IGV) image showing peaks identified in ChIP-seq analysis of *Pb*AP2-G and *Pb*AP2-FG in the upstream region of *pbap2-fg2*.** The grey bar indicates the gene body of *pbap2-fg2*.
(TIF)

**S2 Fig. Genotyping of transgenic parasites developed in this study.** (A) *Pb*AP2-FG2::GFP. (B) *pbap2-fg2*(-). (C) *Pb*AP2-FG2::GFP^C. (D) *Pb*AP2R-2::GFP^C.
(TIF)

**S3 Fig. Alignment of amino acid sequences for previously reported putative AP2 domains for *Pb*AP2R-2 and *Pf*AP2R-2.** The sequences were aligned using the ClustalW program in Mega X. Asterisks indicate amino acids conserved between the two species.
(TIF)

**S4 Fig. Heat maps showing coverage in both IP and input samples for ChIP-seq of *Pb*AP2-FG2 at *Pb*AP2R-2 peaks (left) and for ChIP-seq of *Pb*AP2R-2 at *Pb*AP2-FG2 peaks (right).** Peak regions are aligned in ascending order of their fold enrichment value.
(TIF)

**S1 Table. List of differentially expressed genes in *pbap2-fg2*(-).** (A) RPKM values in each data. (B) Significantly downregulated genes. (C) Significantly upregulated genes.
(XLSX)

**S2 Table. List of sex-enriched genes.** (A) Female-enriched genes. (B) Male-enriched genes.
(XLSX)

**S3 Table. List of peaks and target genes identified in the ChIP-seq experiments of *Pb*AP2-FG2.** (A) Peaks in Experiment 1. (B) Peaks in Experiment 2. (C) Target genes.
(XLSX)

**S4 Table. List of peaks identified in the ChIP-seq experiments of *Pb*AP2R-2.** (A) Peaks in Experiment 1. (B) Peaks in Experiment 2.
(XLSX)

**S5 Table. List of differentially expressed genes in *pbap2r-2*(-).** (A) RPKM values in each data. (B) Significantly downregulated genes. (C) Significantly upregulated genes.
(XLSX)

**S6 Table. List of proteins identified in the RIME using *Pb*AP2-FG2::GFP and WT.**
(XLSX)

**S7 Table. List of peaks and target genes identified in the ChIP-seq experiments of *Py*AP2-O3.** (A) Peaks in Experiment 1. (B) Peaks in Experiment 2. (C) Target genes.
(XLSX)

**S8 Table. List of differentially expressed genes in *pyap2-o3*-null parasite.** (A) FPKM values in each data. (B) Significantly downregulated genes. (C) Significantly upregulated genes.
(XLSX)

**S9 Table. List of primers used in this study.**
(XLSX)

## Author Contributions

**Conceptualization:** Tsubasa Nishi, Masao Yuda.

**Data curation:** Tsubasa Nishi, Izumi Kaneko.

**Formal analysis:** Tsubasa Nishi, Izumi Kaneko, Masao Yuda.

**Funding acquisition:** Tsubasa Nishi, Izumi Kaneko, Masao Yuda.

**Investigation:** Tsubasa Nishi, Izumi Kaneko, Shiroh Iwanaga, Masao Yuda.

**Methodology:** Masao Yuda.

**Project administration:** Masao Yuda.

**Resources:** Masao Yuda.

**Supervision:** Masao Yuda.

**Validation:** Shiroh Iwanaga, Masao Yuda.

**Visualization:** Tsubasa Nishi.

**Writing – original draft:** Tsubasa Nishi, Masao Yuda.

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
