## [Decision Letter · Decision Letter 0]

29 Oct 2022

Dear Dr. Yuda,

Thank you very much for submitting your manuscript "PbAP2-FG2 and AP2R-2 function together as a transcriptional repressor complex essential for Plasmodium female development" for consideration at PLOS Pathogens. As with all papers reviewed by the journal, your manuscript was reviewed by members of the editorial board and by several independent reviewers. In light of the reviews (below this email), we would like to invite the resubmission of a significantly-revised version that takes into account the reviewers' comments.

All reviewers the appreciated the potential advance of the data presented to the field. However they raised major concerns that will required major revisions to answer each of their comments. Primarily, it is necessary to provide experimental data to demonstrate that there is a functional repressor complex which is formed between PbAP2-FG2 and PbAP2R-2.

We cannot make any decision about publication until we have seen the revised manuscript and your response to the reviewers' comments. Your revised manuscript is also likely to be sent to reviewers for further evaluation.

Sincerely,

Ron Dzikowski

Associate Editor

PLOS Pathogens

Kami Kim

Section Editor

PLOS Pathogens

Kasturi Haldar

Editor-in-Chief

PLOS Pathogens

orcid.org/0000-0001-5065-158X

Michael Malim

Editor-in-Chief

PLOS Pathogens

orcid.org/0000-0002-7699-2064

All reviewers the appreciated the potential advance of the data presented to the field. However they raised major concerns that will required major revisions to answer each of their comments. Primarily, it is necessary to provide experimental data to demonstrate that there is a functional repressor complex which is formed between PbAP2-FG2 and PbAP2R-2.

Reviewer's Responses to Questions

**Part I - Summary**

Reviewer #1: Male and female gametocytes are sexual progenitor cells essential for the transmission of the malaria parasite by the mosquito. Differentiation of gametocytes into fertile gametes depends on a sex-specific transcriptional program. How parasites establish different transcriptional repertoires in male and female gametocytes is still largely unknown. In this study, they discovered that the two female transcriptional regulators PbAP2-FG2 and AP2R-2 cooperate as a transcriptional repressor complex in P. berghei, with target genes including male, female, and early gametocyte genes activated by AP2-G. Based on their results, PbAP2-FG2 and AP2R-2 appear to play multiple roles in the development of female gametocytes from early gametocytes. This study was performed in the rodent model of P. berghei malaria and challenges a previous study of PyAP2-O3 (renamed PyAP2-FG2) in the P. yoelii model (reference 18, by Li et al.).

Reviewer #2: In their manuscript: “PbAP2-FG2 and AP2-R2 function together as a transcriptional repressor complex essential for Plasmodium female development”, Nishi et al. characterize the potential transcriptional regulatory role of two ApiAP2, DNA-binidngproteins, AP2R-2 and AP2-FG. Through phenotypic analysis it would seem the AP2-FG2 protein peaks in expression in female gametocytes and the genetic perturbation of this protein results in a decrease in female gametocyte-related genes. The protein seems to exert its function through binding upstream regions of genes, repressing male and female-related gametocyte genes alike. A similar profile was observed for AP2R-2. After thorough reexamination of previously published data for the orthologue of AP2-FG2 in Plasmodium yoelli, the authors demonstrate significant overlap between the function of the two orthologous proteins. Overall, this study confirms that the role of the newly coined “AP2-FG2” in Plasmodium berghei is consistent to what has previously been published for Plasmodium yoelii. I thank the authors for a well organized, and clearly communicated study. However, there are both major and minor concerns with this manuscript and it is requested that the authors respond to each and make edits where appropriate to strengthen the manuscript.

Reviewer #3: The gametocyte stage is a critical step to continuing the Plasmodium lifecycle from the vertebrate host to mosquito vector. However, the fine-tuned regulation of this life stage transition is not completely understood. This manuscript focuses on PbAP2-FG2 and PbAP2R-2, which are two putative female gametocyte regulators. PbAP2-FG2 (for “female gametocyte 2”) was previously named PbAP2-O3 (PBANKA_1015500; “ookinete 3”) in Modryznska et al. (2017) Cell Host & Microbe due to the ap2-o3_KO causing a growth arrest prior to the ookinete stage of mosquito development. In this submission, the authors have renamed the factor to PbAP2-FG2, due to expression in female gametocytes and dysregulation of female gametocyte maturation genes after PbAP2-FG2 disruption. The second factor, PbAP2R-2 (stands for “AP2-related factor 2”) was named in previous work by this group in Yuda et al. (2021) Parasitology International because it contained an ACDC domain and was a target of PbAP2-G and PbAP2-FG via ChIP-seq. In this work, Nishi et al. employed a variety of approaches to identify the role of both PbAP2-FG2 and PbAP2R-2, including (1) GFP-tagging PbAP2-FG2 that identified peak protein expression during female gametocyte development, (2) gene KO of PbAP2-FG2 coupled with RNA-seq that identified an enrichment of female-specific dysregulated transcripts over male-specific transcripts, (3) anti-GFP chromatin immunoprecipitation followed by sequencing (ChIP-seq) of PbAP2-FG2 that identified its genome-wide binding sites and enriched RGAGAR DNA motif, (4) integration of ChIP-seq gene targets and RNA-seq DEGs that identified an enrichment of ChIP-seq targets with the female-specific genes upregulated, (5) CRISPR-Cas9 modifications to remove the RGAGAR motif from a promoter region that identified a depletion of PbAP2-FG2 binding by ChIP-qPCR, (6) CRISPR-Cas9 to integrate a GFP tag onto the 3`-end of pbap2r-2 and an anti-GFP ChIP-seq that identified the binding sites of PbAP2R-2, which had a high overlap with PbAP2-FG2 ChIP-seq binding sites, (7) integration of PbAP2-G ChIP-seq targets that identified many gene targets of PbAP2-G as targets of PbAP2-FG2 and PbAP2R-2, and finally (8) a re-analysis of ChIP-seq data from Li et al. (2021) EMBO Reports on the PbAP2-FG2 P. yoelii ortholog, PyAP2-O3, where they identified their enriched RGAGAR motif in the highly reproducible ChIP-seq binding sites.

Overall, the results presented in this manuscript regarding PbAP2-FG2 are exciting and the data analysis is thorough. The results provide a new characterization of an ApiAP2 protein PbAP2-FG2 involved in the regulation of gametocyte development, particularly in female cell fate determination. However, despite a number of preliminary results, the suggestion that the data presented demonstrate a transcriptional regulatory complex between PbAP2-FG2 and PbAP2R-2 that work together to repress gene targets is unsubstantiated. The data clearly show that PbAP2-FG2 and PbAP2R-2 target highly similar genomic regions, but proof that those components are physically interacting in the same cell at the same time are lacking.

**Part II – Major Issues: Key Experiments Required for Acceptance**

Reviewer #1: 1- I found quite puzzling that there few if no correlation between the direct targets of PbAP2-FG2 (assessed by ChIP-seq) and the dysregulation of the transcriptome in its absence. The authors have attempted to explain these discrepancies, but their hypothesis is elusive and does not allow for the possibility that this factor could be replaced, for example, by another ApiAP2. Is it possible that each AP2 they are describing homodimerizes individually and compensates for the absence of the other.

2- With their elegant experiments on cis-mutagenesis of the repressive element, they provide only a partial answer to the lack of correlation at the genomic level between ChIPseq and RNAseq by proposing that PbAP2-FG2 and PbAP2-R2 cooperatively repress genes, probably as heterodimers. However, the possibility that both homodimerize and colocalize at the same genomic sites cannot be ruled out until it is shown that they interact biochemically or that PbAP2-R2 is released from chromatin in the absence of PbAP2-FG2 and vice versa.

Reviewer #2: 1. Secondarily, the authors abandon any notion that this protein has a role in any other stage of development quite quickly, despite it’s obvious expression in both male gametocytes and ookinetes. Could the authors clarify why neither of these were pursued further? Is it clear to them that there is no possibility of a role for this AP2 in other stages of development, why or why not? In addition, please quantify all of the phenotypes that are suggested by both the GFP-tagged and genetic disruptants of AP2-FG2

2. The authors have completed a tremendous amount of work to characterize these proteins and it will add to the already large number of ApiAP2s this lab has revealed the role of. They are to be commended for their continued efforts. However, to support the assertion that AP2-FG2 and AP2R-2 act together to co-repress genes, the authors must provide definitive data that demonstrate an interaction between the two lines. The authors should consider Co-IP or tagging AP2-FG2 with 3xHA in a parasite that has AP2R-2 tagged with GFP. Flow-cytometry can be used to select for parasites expressing AP2R-2 and ChiP-seq for either or both AP2 can be performed.

Reviewer #3: There is insufficient data presented to demonstrate that there is a complex formed between PbAP2-FG2 and PbAP2R-2 implying colocalization. Therefore, the title and many statements in the Results and the beginning and end of the Discussion need to be modified to reflect this. Although the authors do demonstrate that the PbAP2-FG2 and PbAP2R-2 appear to occupy similar genomewide binding sites, they do not test whether this is due to complex formation. At least two possibilities come to mind. It is entirely feasible that in 2 different cell populations PbAP2-FG2 is binding these loci throughout the genome, while in a different population PbAP2R-2 is binding the same loci. This can only be resolved by single cell ChIP-seq which has not been successful to date in Plasmodium parasites (to my knowledge). Although this is unlikely given the roughly 1:1 correlation of the ChIP-seq peak intensities reported, it is possible. To directly test whether there is a complex, immunoprecipitation (IP) experiments will need to be performed. Both PbAP2-FG2 and PbAP2R-2 were tagged in this manuscript and used for localization studies and ChIP-seq. Therefore, IP-MS proteomics should be feasible in principle. To claim that PbAP2-FG2 and PbAP2R-2 form a complex, the authors should produce IP/MS data using all available tagged reagents already at their disposal.

Throughout the manuscript (e.g. Line 73, Line 533), reference to P. berghei targets of AP2-G are mentioned solely in the context of previous work by the same authors, However, other studies have determined targets of PfAP2-G by inducible activation, ChIP-seq, or scRNA-seq including Llora-Battle et al. 2019, Josling et al. 2020, and Poran 2017. Two points are worth addressing. First, what is the overlap in predicted targets between these studies? Can the authors speculate why they are different between PbAP2-G (many targets) and PfAP2-G (limited target set)? For example, the authors previously reported pbap2-g2, pbap2-fg, and pbab2-z as targets of PbAP2-G. However, according to the literature this is not the case for PfAP2-G, where these other ApiAP2 genes are not direct targets.

Next, overall the manuscript is highly under-cited. Please be more inclusive in recognizing the contributions of others. One major omission in the Introduction is reference to the recent work of Russell et al. who have clearly demonstrated roles for a helix-turn-helix protein and other zinc-finger proteins in the regulation of male and female sexual development lineages of Pb parasites: BioRxiv: 2021.08.04.455056v1 This study is highly relevant to the current work and an understanding of sexual maturation and sex lineage differentiation.

Given these potential species-specific differences, it would be worthwhile to identify each ApiAP2 discussed with the relevant species designation (Pb, Py, Pf) based on where the evidence presented originates. There are clear reasons to believe that target genes in one organism may or may not be the same across species. Similarly, it has already been demonstrated that certain homologous ApiAP2 proteins are essential in some species, but not all. By definition, their functional roles must therefore be different or there is functional redundancy in one species. Therefore, please use full IDs throughout such as PbAP2-G2 or PbAP2R-2 or PfAP2-G2, etc…

Another clear issue is comparing the results in lines 389-294 between Pf and Pb and the role of the cytoskeleton. Clearly the timing and duration of these complexes must be vastly different given the 24-hour maturation of gametocytes in Pb versus 10 days in Pf.

In generating tagged parasite lines for this study (Figure 1C) it is unclear if the endogenous 3`UTR is maintained in these parasites. This same group reported that altering the endogenous 3’ UTR was found to be an issue in previous work on AP2-Z (Yuda et al. (2021) and Nishi et al. (2022) BioRxiv). This was reportedly because the replacement of the endogenous 3`UTR disrupted the normal function of DOZI and the translational repression of AP2-Z. Please label the plasmid designs in Fig S1 more clearly and please note any changes to the 3`UTR in the main text.

The authors use a nice cross-fertilization assay to demonstrate the role of pbap2-fg2. The same should be done for pbap2r-2, and the tools to do so are available.

To determine differentially expressed genes (DEGs), the authors reference “previously reported sex-specific RNA-seq data” from reference 21 (lines 168, 169). This choice appears arbitrary. There are numerous datasets available (transcriptomic, proteomic, genomic) that have catalogued putative male and female gametocyte markers. A much better approach would be to use a comprehensive table with multiple lines of evidence to categorize individual genes as “male” or “female”, especially since most of them have not been independently characterized. Further, these genes should be referred to as “male-enriched” and “female-enriched” as opposed to “male-specific” or “female-specific”, because few genes are only truly present in one lineage.

The ChIP-seq results are quite surprising with over 1000 peaks in every replicate experiment. Compared to other ChIP-seq results in the Plasmodium literature, this number is very high for an individual DNA binding protein. Looking back, virtual all papers from this group report high numbers of binding events. What number of genes does this represent? Presumably not 1/5 of the genome? If not, does the appearance of multiple binding events/motifs upstream of genes suggest something associated with regulation?

Due to the high overlap in binding sites between PbAP2-FG2 and PbAP2R-2, the authors should check whether those sites are also pulled down when performing an anti-GFP ChIP-seq in a WT parasite background (i.e., are those sites hyper ChIP-able)?

Also, how many of the more than one thousand binding sites have the motif reported in Figure 3C. It is not clear. Please prepare a figure showing the peak summits for all binding events colored by the underlying nucleotide sequence.

The statement in line 320-321 that AP2-R2 has no AP2 domain is unsubstantiated. What is this evidence based on? The few descriptions in the literature of the ACDC domain suggest that it must always co-occur with an AP2 domain. In fact, in the Pf orthologue (PF3D7_1319600), Oehring et al. 2012 (Figure 8) identified 2 AP2 domains in this “ACDC domain-containing protein”. Please provide an alignment in the Supplementary materials between the Pb and Pf orthologues for AP2R-2. Also, using AlphaFold2 to examine the predicted structure of AP2R-2 clearly demonstrates at least one well-folded AP2 domain and perhaps a cryptic second AP2 domain. The authors should provide this information. In addition to demonstrating that this protein may have 1-2 AP2 domains, it may also provide an alternative interpretation of why the binding of AP2R-2 and AP2-FG2 are so similar? Perhaps their AP2 domains are actually relatively similar and therefore bind the same motif and genome-wide sites? Otherwise, it is also unclear why the authors pursue ChIP-seq of AP2R-2 if they do not suspect that it has any DNA binding domains.

In the re-analysis of the PyAP2-FG2 ChIP-seq data from Li et al. please describe clearly how your methods of analyzing the ChIP-seq data differ. You describe how you have re-analyzed their data, but have you re-analyzed it using their approach? And which approach is better and why beside the inclusion/exclusion of low read counts?

Renaming genes is a tricky business. Is there sufficient data here to fully support this? I’m not sure. I certainly do not think it is a good idea to continue renaming ApiAP2 transcription factors? This will lead to serious confusion in the field. One simple reason is that it is possible that they may serve various functions throughout the entire life cycle, so will each gene product get a different name every time a new phenotype is published? Also, species-specific differences should be anticipated given differences in the lifecycles. Within each publication it is easy to follow which gene is being discussed, but when comparing across publications it will become increasingly difficult to make comparisons. For example in lines 94, 98, and 100: you call the P. yoelii ortholog “PyAP2-FG2” but Li et al. technically call it “PyAP2-O3”. It should be PyAP2-O3 to be in-line with Li et al. and then “PbAP2-FG2” for your publication.

**Part III – Minor Issues: Editorial and Data Presentation Modifications**

Reviewer #1: 1- The authors are advised to follow the nomenclature that precedes their work and not to change the genes already named by their colleagues to suit their ideas. Li et al (reference 18) used PyAP2-O3 in their article, which has been changed to PyAP2-FG2 (lines 93-95) in this manuscript, with the authors stating that PyAP2-O3 is the bona fide ortholog of PbAP2-FG2. The reviewer found this very confusing.

2- The authors compared their data with those of Li et al (reference 18). They reanalyzed all omic data, starting with the raw data deposited by Li et al, and reinterpreted their data in a very parsimonious way, concluding that they were wrong, and finally proposing their model as the true model. In my opinion, it would be beneficial if the authors were less direct in their criticisms (perhaps they should curb their enthusiasm in "killing" a competing story) and leave room for alternative hypotheses. I suggest rewriting this part with less "enthusiasm," I would say.

3- Missing scales in Figures :

• y-axis scale in figures 1B, 7B

• bar scale in figuress 1C-E

4- Figure 1D : The parental strain must be shown in parallel to demonstrate that there is no difference in the morphology of gametocytes produced by a mutant pbap2-fg2(-) background.

5- ChIPseq figures 3A and 6A : Rather than plotting a genomic area, the authors should compare enrichment levels across the genome by plotting pairwise correlations between replicates (i.e., plotting the average tag number of rep_1 versus rep_2 enrichment).

Reviewer #2: 1. The authors pursue the characterization of PBANKA_1015500, which contains both an ApiAP2 and an ACDC domain. This protein has previously been characterized by Li et al. 2021 (PMID: 33665945) as AP2-O3. Here the authors chose to re-name this protein to AP2-FG2. Little has been done to prove this protein has no role in male gametocytes or other stages of development. Since there is no official naming convention, anyone can name them based on the role of these proteins in various stages and species. Have the authors considered not renaming the protein for consistency in the field?

2. In Figure 1B, the IFAs used to substantiate the lifecycle stage-specific expression of AP2-FG2 are very difficult to discern. However, the quality of the image makes it very hard to properly assess this claim. The authors should support this assertion via an orthogonal method, such as Western Blot analysis at every stage of development. In addition, the authors should quantify their claim of stage-specific expression. Please quantify the percentage of males and females that have GFP expression.

3. The authors pursue the characterization of PBANKA_1015500, which contains both an ApiAP2 and an ACDC domain. This protein has previously been characterized by Li et al. 2021 (PMID: 33665945) as AP2-O3. Here the authors chose to re-name this protein to AP2-FG2. Little has been done to prove this protein has no role in male gametocytes or other stages of development. Since there is no official naming convention, anyone can name them based on the role of these proteins in various stages and species. Have the authors considered not renaming the protein for consistency in the field?

4. In line 128-129 the authors state two clonal lines were produced but never clarify which of the clones were used in each experiment, with the exception of their mosquito feeding experiments in line 135-136.

5. Are the images in Figure 1D and E representative images? Did the authors quantify morphologically normal/abnormal gametocytes and ookinetes compared to the WT parent? This developmental phenotype should be quantified for each knockout clone. Additionally, the WT representative images should be shown for comparison (Figure 1D).

6. In line 131-132 the authors assert that the ap2-fg2 knock out line produces normal amounts of exflagellated parasites but do not provide any supporting data. Please quantify the number of exflagellation events for both clones.

7. Line 135-136 the authors also do not show data to support the assertion that there is a loss of ookinete production in the ap2-fg2 knock out line (either clone).

8. Figure 1F is missing from the manuscript file, the authors should also state how many biological and technical replicates were performed for the experiments whose data is presented in these panels.

9. The authors should clarify the remark, “overall, log2(fold change) of female-enriched genes tended to be lower than the other genes” (line 174-176). Is this in context of the significantly decreased transcripts only or the total transcriptome? Comparing the expression of these classes of genes with a two-tailed t-test is also not strictly appropriate, as there would be thousands of genes falling into the other category that are not significantly different between WT and KO. Perhaps the more appropriate test would be to perform a Fisher’s exact test to check for relative overrepresentation of female-enriched genes in the significantly decreased transcripts identified in the study.

10. The authors determined the genomic location of their called peaks from the ChIP-seq. Out of the almost 3,000 peaks from both experiments, what was the distribution of the peaks in intergenic vs intragenic regions? For those that did not have a recognized motif, was the distribution different?

11. The authors postulate the role of AP2-FG2 as a transcriptional repressor and the authors note in Table S3 whether or not each potential target gene is significantly up or down regulated in the AP2-FG2(-) transcriptomics. To assess the significance, the authors should present the distribution (not necessarily significantly up or down) of all the identified ChIP target genes in the transcriptomics experiments.

12. Line 308-311: To demonstrate sex-specificity using the RT-qPCR, the authors failed to use a published Why did the authors use a male and female marker to compare the RT-qPCR results to instead of a published reference gene?

13. The authors also assert in their heading line 315-316 that AP2-FG2 requires a corepressor to repress its target genes, but the authors do not show that the binding of AP2-FG2 is lost in the absence of AP2R-2.

14. The authors claim that the genome-wide distribution of AP2-FG2 and AP2R-2 are highly similar based on data shown in Figure 6A and B (Lines 329-330). However, Figure 6A shows only a small portion of the genome and Figure 6B shows read coverage data instead of log2 fold enrichment. It would be more convincing if the authors instead showed the overlapping peaks the way they show for the overlap with AP2-G (figure 7A) or alternatively use the IDR1D analysis.

15. The way the authors showed the overlap between the increased and decreased transcripts between AP2-FG2 and AP2R-2 in Figure 6E with a correlation plot is not appropriate for the distribution of the data. For a linear correlation in the data to be drawn, the data needs to follow a linear distribution, which is not the case for their increased and decreased transcripts. The authors can simply show what proportion of transcripts are differentially abundant in each of the conditions.

16. Please scale ChIP-seq tracks between groups and include y- and x-axes on ChIP-seq tracks for figures 1,3,6 and 7. When showing ChIP-seq tracks also clarify whether what is being displayed is truly “coverage” or actually log2(ChIP/input).

17. In Supplementary Figure 1: the authors should state which primer sets illustrated in the pictograms produced the PCR products shown in the figure and the conditions of the PCR should be stated: nr of cycles, annealing temperature of primers should be included in supplementary table S8.

18. Line 455-457: The authors state they set their FPKM threshold at <10, but it seems they meant to include transcripts with FPKM >10.

19. Please revise or complete the following requests for the Materials and Methods section of the paper:

a. Please include a detailed methodfor the “fluorescence analysis” of the parasites- including what microscope was used, if this was an immunofluorescence assay or live fluorescence microscopy etc.

b. For the generation of plasmid for transgenic alteration of the parasite, include more details on the plasmid background, with appropriate references, should be stated clearly, with a higher level of detail than simply naming them: ie. “gfp-fusion vector” and “targeting construct”

c. The authors stated the parasite lines were verified by PCR and/or Sanger sequencing. Please specifically state which methods were used in which cases, if Sanger sequencing was used to verify these lines please provide supplementary data to this effect.

d. The authors should consider performing next-generation whole genome sequencing on the knockout parasite lines to verify that there are no other mutations that could contribute to the phenotype.

e. For the ChIP-seq experiments- there seems to be substantial information about the experimental protocol missing without any supplementary method being cited. Please provide a more detailed protocol and ensure to include the following detail OR cite the source of the protocol that includes these details: Was a nuclear extraction performed prior to sonication? What is the composition of the SDS lysis buffer? At which temperature were the samples sonicated? What was the concentration of anti-GFP antibodies used in the ChIP-seq experiment and how long was the antibody incubation performed? Were the lysates precleared before the immunoprecipitation step? When were the input control samples taken? The authors also don’t mention a proteinase digestion step, was this omitted? How was the immunoprecipitated DNA purified?

f. There is also information missing regarding the data analysis of the ChIP-seq experiments: What version of the Plasmodium berghei genome was used to map the sequencing reads? How/why did the authors decide on a fold change enrichment threshold of >3?

g. For the ChIP-qPCR and RT-qPCR experiments, what were the starting concentrations of DNA or RNA included in the experiments respectively and how many cycles were these experiments conducted for? Did the authors include reference genes for either study for normalization? No account is given of how these data were analyzed, was the 2-delta Ct method used for quantification in the qPCR?

20. Why did the authors use the Fisher’s exact test to analyze enriched motifs in 6 bp bins instead of using available softwares (ie STREME) that scan over a wider range? Could you please justify your use of the Fisher’s exact test in this context?

21. How did the authors associate genes with Chip-seq peaks if the binding site overlaps within 1200 bp of the start codon? This method does not take into account which genes are closest to the binding site or whether these bound regions are intra- or intergenic.

Reviewer #3: In general, please check the entire document for English grammar. There are examples beginning with the second sentence of the Abstract that require further attention.

The PlasmoDB IDs for all genes mentioned in the manuscript should be provided, especially since the authors are choosing to rename pbap2-o3 as pbap2-fg2. (Such as line 317 and others throughout.)

The word “identical” is used throughout the manuscript. All instances should be replaced with “highly similar” or “analogous”, etc… In biology virtually nothing is identical.

Line 68 – Probably want to use “ability” instead of “capability”.

Please be consistent in the use of AP2R-2 instead of ap2-R2 (see lines 84 and 86). Please correct throughout the manuscript.

The last section of the Introduction (lines 96-100) should be rewritten to simply state that a re-analysis of a previous PyAP2-FG2 dataset leads to similar conclusions despite being at odds with what was previously reported in the Py study.

Lone 110-111, it is unnecessary to state what a previous study failed to do. Alternatively, they may suggest that a gene may have an earlier role leading up to the point of the observed phenotype used to name ap2 genes earlier (by Modrzynska and others). This is especially true when in lines 135-137 the authors compare their results in a confirmatory manner with this previous study.

On line 145, please clarify why 30% conversion (and not 50%) to banana-shaped ookinetes is expected?

Please clarify the sentence on lines 224-226: “The target genes contained some groups of genes seemingly expressed in female gametocytes…” What does it mean to be “seemingly expressed”?

Again, on lines 244-234, the authors write: “…the number of male-enriched genes in the targets seemed too large, again implying the possible role of PbAP2-FG2 as a transcriptional repressor.” You cannot write “seemed too large” unless you have a comparative analysis? On what basis do the authors make this claim?

In lines 248-249, the authors should note that it is very difficult to distinguish the direct vs. indirect targets by RNA-seq analysis alone, therefore target genes need to be identified within the DEGs.

The candidate gene psh3 used the section beginning on lines 277-278 was selected due to its expression and essentiality in P. falciparum parasites. Is this essentiality also true in P. berghei (PlasmoGEM or other datasets?). Also, doesn’t this imply that any regulation via PbAP2-FG2 must be exclusive to the sexual stage and that the expression in trophozoites and schizonts must be via a different factor in the asexual blood stages? I suppose that I’m not convinced as to why psh3 was used for this validation? Why not use a gene that has been well-characterized in Pb?

Line 317, please spell “Previously” correctly.

Line 348, delete “were”.

Line 362, “approximately 1” should be 0.97.

Line 365-366 should read: “Therefore we hypothesized that AP2R-2 may function as an essential co-repressor of PbAP2-FG2 in female gametocytes.”

Lines 371-372 are unclear. What “changes” is not well defined.

In lines 376-378 it is reported that the targets of AP2-G (Pb I presume?) and PbAP2-FG2 overlap, but what about the mRNA abundance in the knockouts. Does the abundance go up (activated, or loss of repression) after PbAP2-FG2 knockout? Please clarify.

Line 414, do not use “implausible”. Please replace with “unlikely”.

In line 435, how is it possible that only 271 of 781 targets genes have orthologues between Py and Pb? These organisms are remarkably similar overall. Please clarify.

On line 440, you state that “…some targets might have been falsely detected.” It is not clear how a target can be falsely detected. Surely based on the analysis used by Li et al. they did measure a target peak.

In lines 440-441, how did you “…assess the sex-specific expression of the target genes of PyAP2-FG2…”?

You should not write on lines 450-451 “… yield unreliable DEG lists.” This is demeaning to the previous study. Perhaps you mean they “… yield much larger DEG lists with larger variance.” Or something along those lines?

Line 455 change to “we re-analyzed”.

How many orthologues were considered in line 461?

Line 45574 change to “the majority”.

Figure comments:

I believe that the Figure 1B ChIP-seq data is published data and should not be in a main figure. Perhaps a supplementary figure?

Figure 1: Is the IGV image group auto scaled? Meaning is the y-axis consistent between both tracks? The y-axis ranges should be indicated on the figure or in the caption.

Figure 1F: This panel regarding the crossing experiment where you have crossed a female and a male gametocyte infertile lineage with ap2-fg2 (-) is missing. Please add this.

In Figure 2 B and C, are the other genes not assigned to male and female still gametocyte genes? Or genes expressed during other (multiple perhaps?) stages? Please clarify this in the manuscript and figure legend. The same question applies to Figure 3G.

In Figure 4, the majority of the differences are really not significant. With a reduced number of candidates, it is not surprising that motif enrichment is stronger. Also, can you explain the data that emanate from the bottom of your plots in the bottom right quadrants in Figure 4A that form a line not congruent with the rest of the volcano plot?

Figure 6B: As you know, it is more widely accepted to show ChIP-seq data as a Log2 fold enrichment of the IP over the input. Why are the data shown as read coverage? From the methods, you demonstrate that you took an input sample, so it would be a nice control to show the heatmaps with the input data as the Fold Enrichment (Log2[IP/Input]), instead of just read coverage.

In Figure 8A, I would consider switching the 2nd and 3rd columns to make your point more strongly to the reader since the p-value is higher for the GAGA motif.

In Supplemental Figure S1, please label the genotyping PCR gels more clearly so that it is clear which primers are being used to generate each product. Also it would be helpful to have the parasite line name (i.e., PbAP2-FG2::GFP) above each schematic in this figure. Is the 3`UTR modified in each of these lines?

PLOS authors have the option to publish the peer review history of their article (what does this mean?). If published, this will include your full peer review and any attached files.

Reviewer #1: No

Reviewer #2: No

Reviewer #3: No
---

## [Decision Letter · Decision Letter 1]

26 Jan 2023

Dear Dr. Yuda,

Thank you very much for submitting your manuscript "PbAP2-FG2 and PbAP2R-2 function together as a transcriptional repressor complex essential for Plasmodium female development" for consideration at PLOS Pathogens. As with all papers reviewed by the journal, your manuscript was reviewed by members of the editorial board and by several independent reviewers. The reviewers appreciated the attention to an important topic. Based on the reviews, we are likely to accept this manuscript for publication, providing that you modify the manuscript according to the review recommendations.

Two of the reviewers pointed out that a GFP tagged unrelated line should be more appropriate negative control for unspecific interactions between the two AP2 proteins. Please address this concern in you revised version in addition to their additional comments.

Sincerely,

Ron Dzikowski

Academic Editor

PLOS Pathogens

Kami Kim

Section Editor

PLOS Pathogens

Kasturi Haldar

Editor-in-Chief

PLOS Pathogens

orcid.org/0000-0001-5065-158X

Michael Malim

Editor-in-Chief

PLOS Pathogens

orcid.org/0000-0002-7699-2064

Two of the reviewers pointed out that a GFP tagged unrelated line should be more appropriate negative control for unspecific interactions between the two AP2 proteins. Please address this concern in you revised version in addition to their additional comments.

Reviewer Comments (if any, and for reference):

Reviewer's Responses to Questions

**Part I - Summary**

Reviewer #1: This is a timely and well-done study. The authors responded to all the points this reviewer raised. The paper was significantly improved with the addition of new controls and complementary experiences.

Reviewer #2: In their manuscript: “PbAP2-FG2 and AP2-R2 function together as a transcriptional repressor complex essential for Plasmodium female development”, Nishi et al. characterize the potential transcriptional regulatory role of two ApiAP2,

DNA-binidngproteins, AP2R-2 and AP2-FG. Through phenotypic analysis it would seem the AP2-FG2 protein peaks in expression in female gametocytes and the genetic perturbation of this protein results in a decrease in female gametocyte-related genes. The protein seems to exert its function through binding upstream regions of genes,

repressing male and female-related gametocyte genes alike.

The authors were given review comments and I appreciate the authors efforts to address the concerns I had with their study following the initial submission. Most comments/concerns were adequately addressed and the addition of RIME to determine the interacting partners of AP2-FG2 strengthens this study. I am, however, still curious about the underlying AP2-FG2 GFP-tagged line. I have commented on the authors responses in line in the attached document and summarized below in the "Minor Issues" section

Reviewer #3: Overall the authors have addressed all of my major concerns. I have some additional suggestions and minor points raised below.

**Part II – Major Issues: Key Experiments Required for Acceptance**

Reviewer #1: (No Response)

Reviewer #2: (No Response)

Reviewer #3: (No Response)

**Part III – Minor Issues: Editorial and Data Presentation Modifications**

Reviewer #1: (No Response)

Reviewer #2: In response to Reviewer #2 Comment 2, the authors suggest that the GFP-signal does not need to be quantified or more clearly demonstrated to be exclusive to female gametocytes. I think it is good scientific practice to confirm any imaging with Western Blot analysis. Part of establishing confidence in the genetic line that has been created and reagents used in the ChIP-seq is first proving that the full-length GFP-tagged PbAP2-FG2 protein is exclusively expressed in the female gametocyte and that it is cleanly identified by the antibody used for the ChIP-seq by showing a Western Blot.

In response to Comment 13, I thank the authors for performing the additional RIME experiment that empirically establishes the interaction of PbAP2-FG2 and PbAP2R-2. Could the authors speculate as to why AP2-I, AP2-G2 and AP2-O2 are also found to complex with PbAP2-FG2? This should be added to the discussion of the results of RIME. Additionally, please discuss your results with respect to the recently published P. berghei male/female gametoycte differentiation and maturation ssRNA-seq study https://doi.org/10.1016/j.chom.2022.12.011.

In response to Comment 14, I thank you for the adding the Venn diagram which shows an overlap of peaks between the two ChIP-seq experiments. While we understand your justification for presentation of data, if your data analysis strictly follows ENCODE guidelines (for coverage, quality, and reproducibility), visual analysis of the quality of IP experiments should not be necessary. If you prefer to show read coverage of your IP samples, please include the read coverage from the corresponding input in each panel so that the quality of the IP experiments can be directly evaluated as you suggest.

In response to Comment 15, thank you for attempting to address our initial comment regarding the correlation plot displayed as Figure 6H. WI apologize for not being clearer in the original request. There are a couple of reasons that this analysis is statistically inappropriate. 1) The data points on the graph are not calculated entirely independent of each other. For the FC calculation, you do not have an independent WT measurement for each knockout sample. Each WT timepoint is used in the calculation of one of each knockout and, therefore, the Log2FC calculations are not completely independent of each other, and linear regression/approximation is not appropriate. (For example: What you are currently doing is A/B to C/B, where A and C are your knockout strains and B is the WT). The correct comparison would be A/B to C/D OR directly A to C. 2) The data are already filtered for statistical significance resulting in two “clusters” of data points and skewing the data toward a linear relationship based on the extreme fold changes.

There are a few things that can be done to make your analysis appropriate:

1) To draw a statistical conclusion from a correlation plot all of the data should be independent of each other. For instance, since your denominator for the Log2FC calculation per gene is the same across the knockout genotypes, you might consider just plotting the average expression value per gene with each knockout represented on a different axis, followed by a linear regression/approximation analysis. One can use a density correlation plot for large data sets.

2) Alternatively, the Log2 expression of significant genes in either knockout line can be represented on separate graphs as a box plot where the WT expression of select significant genes is separate from your knockout expression and the upward or downward trend will support/refute the hypothesis.

3) More simplistically, you could represent the significantly up or downward FC of sets of genes in both knockout strains as a Venn Diagram to demonstrate the overlap of regulated genes.

Reviewer #3: In the revised manuscript the authors now provide evidence for a direct interaction between PbAP2-FG2 and PbAP2-R2 using Rapid Immunoprecipitation Mass Spectrometry of Endogenous Proteins (RIME), which combined ChIP and MS to identify associated proteins. Although using an unrelated GFP-tagged line to monitor unspecific interactions would have been a more appropriate negative control for this experiment, the pull-down of PbAP2-FG2 detected PbAP2-R2 demonstrating that PbAP2-FG2 and PbAP2-R2 likely interact. Other proteins shown in Figure 7 appear to be more hand selected from their IP data (based on examination of Table S6), and I am less convinced by the results based solely on enrichment (over the WT control), such as PbMORC and other chromatin remodeling complexes, although these are interesting observations.

A few additional minor points:

Line 21 of the abstract should read: “Despite the number of studies on gametocyte development that have been conducted, the molecular mechanisms regulating this process remain to be fully understood.”

If PbAP2-FG2 and PbAP2R-2 function together, why does the abstract only mention “a significant overlap between the target genes of PbAP2-FG2 and AP2-G.” Why not also add PbAP2R-2 since these two proteins overlap in target gene specificity?

Should PbAP2R-2 be renamed PbAP2-FG3? Why or why not?

Line 111, please change “AP2-family transcription factor” to “ApiAP2-family transcription factor”

The authors should modify their citation of Russell et al. since it has now been published in Cell Host & Microbe and is no longer solely a bioRxiv preprint.

The authors have added additional support for the presence of an AP2 domain (at least one) in PbAP2-R2 (lines 340-346). Please include alignments of the regions that they find to not be well-conserved between Pb and Pf as supplemental data. Also, I would again encourage the authors to use AlphaFold2, which clearly demonstrates at least one well-folded AP2 domain. Given that this protein is likely to contain at least one AP2 domain, it would be good to speculate what the role of this AP2 domain might be given that the binding matches that of PbAP2-FG2?

Finally, I have an alternative title suggestion: “PbAP2-FG2 and AP2R-2 function together as an essential transcriptional repressor complex during female gametocyte development in Plasmodium”

PLOS authors have the option to publish the peer review history of their article (what does this mean?). If published, this will include your full peer review and any attached files.

Reviewer #1: **Yes: **HAKIMI Mohamed-Ali

Reviewer #2: No

Reviewer #3: No

Figure Files:

Data Requirements:

Reproducibility:

References:

---

## [Editor Report · Decision Letter 2]

2 Feb 2023

Dear Dr. Yuda,

We are pleased to inform you that your manuscript 'PbAP2-FG2 and PbAP2R-2 function together as a transcriptional repressor complex essential for Plasmodium female development' has been provisionally accepted for publication in PLOS Pathogens.

Best regards,

Ron Dzikowski

Academic Editor

PLOS Pathogens

Kami Kim

Section Editor

PLOS Pathogens

Kasturi Haldar

Editor-in-Chief

PLOS Pathogens

orcid.org/0000-0001-5065-158X

Michael Malim

Editor-in-Chief

PLOS Pathogens

orcid.org/0000-0002-7699-2064
---

## [Editor Report · Acceptance letter]

7 Feb 2023

Dear Dr. Yuda,

We are delighted to inform you that your manuscript, "PbAP2-FG2 and PbAP2R-2 function together as a transcriptional repressor complex essential for Plasmodium female development," has been formally accepted for publication in PLOS Pathogens.

Best regards,

Kasturi Haldar

Editor-in-Chief

PLOS Pathogens

orcid.org/0000-0001-5065-158X

Michael Malim

Editor-in-Chief

PLOS Pathogens

orcid.org/0000-0002-7699-2064